# Fast Optimal Transport through Sliced Wasserstein Generalized Geodesics

**Guillaume Mahey**
INSA Rouen Normandie - Université Bretagne Sud
LITIS - IRISA
`guillaume.mahey@insa-rouen.fr`

**Laetitia Chapel**
Université Bretagne Sud - Institut Agro Rennes-Angers
IRISA
`laetitia.chapel@irisa.fr`

**Gilles Gasso**
INSA Rouen Normandie
LITIS
`gilles.gasso@insa-rouen.fr`

**Clément Bonet**
Université Bretagne Sud
LMBA
`clement.bonet@univ-ubs.fr`

**Nicolas Courty**
Université Bretagne Sud
IRISA
`nicolas.courty@univ-ubs.fr`

## Abstract

Wasserstein distance (WD) and the associated optimal transport plan have proven useful in many applications where probability measures are at stake. In this paper, we propose a new proxy for the squared WD, coined min-SWGG, which relies on the transport map induced by an optimal one-dimensional projection of the two input distributions. We draw connections between min-SWGG and Wasserstein generalized geodesics with a pivot measure supported on a line. We notably provide a new closed form of the Wasserstein distance in the particular case where one of the distributions is supported on a line, allowing us to derive a fast computational scheme that is amenable to gradient descent optimization. We show that min-SWGG is an upper bound of WD and that it has a complexity similar to that of Sliced-Wasserstein, with the additional feature of providing an associated transport plan. We also investigate some theoretical properties such as metricity, weak convergence, computational and topological properties. Empirical evidences support the benefits of min-SWGG in various contexts, from gradient flows, shape matching and image colorization, among others.

## 1 Introduction

Gaspard Monge, in his seminal work on Optimal Transport (OT) [42], studied the following problem: how to move with minimum cost the probability mass of a source measure to a target one, for a given transfer cost function? At the heart of OT is the optimal map that describes the optimal displacement as the Monge problem can be reformulated as an assignment problem. It has been relaxed by [33] by finding a plan that describes the amount of mass moving from the source to the target. Beyond this optimal plan, an interest of OT is that it defines a distance between probability measures: the Wasserstein distance (WD).

Recently, OT has been successfully employed in a wide range of machine learning applications, in which the Wasserstein distance is estimated from the data, such as supervised learning [30], natural

language processing [38] or generative modelling [5]. Its capacity to provide meaningful distances between empirical distributions is at the core of distance-based algorithms such as kernel-based methods [60] or $k$-nearest neighbors [6]. The optimal transport plan has also been used successfully in many applications where a matching between empirical samples is sought such as color transfer [55], domain adaptation [19] and positive-unlabeled learning [15].

Solving the OT problem is computationally intensive; the most common algorithmic tools to solve the discrete OT problem are borrowed from combinatorial optimization and linear programming, leading to a cubic complexity with the number of samples that prevents its use in large scale applications [53]. To reduce the computation burden, regularizing the OT problem with e.g. an entropic term has led to solvers with a quadratic complexity [23]. Other methods based on the existence of a closed form of OT have also been devised to efficiently compute a proxy for WD, as outlined below.

**Projections-based OT.** The Sliced-Wasserstein distance (SWD) [56, 10] leverages 1D-projections of distributions to provide a lower approximation of the Wasserstein distance, relying on the closed form of OT for 1D probability distributions. Computation of SWD leads to a linearithmic time complexity. While SWD averages WDs computed over several 1D projections, max-SWD [24] keeps only the most informative projection. These frameworks provide efficient algorithms that can handle millions of samples and have similar topological properties as WD [45]. Other works restrain SWD and max-SWD to projections onto low dimensional subspaces [52, 40] to provide more robust estimation of those OT metrics. Although effective as proxies for WD, those methods do not provide a transport plan in the original space $\mathbb{R}^d$. To overcome this limitation, [44] aims to compute transport plans in a subspace which are extrapolated to the original space.

**Pivot measure-based OT.** Other research works rely on a pivot, yet intermediate measure. They decompose the OT metric into Wasserstein distances between each input measure and the considered pivot measure. They exhibit better properties such as statistical sample complexity or computational efficiency [29, 65]. Even though the OT problems are split, they are still expensive when dealing with large sample size distributions, notably when only two distributions are involved.

**Contributions.** We introduce a new proxy for the squared WD that exploits the principles of aforementioned approximations of OT metric. The original idea is to rely on projections and one-dimensional assignment of the projected distributions to compute the new proxy. The approach is well-grounded as it hinges on the notion of Wasserstein generalized geodesics [4] with pivot measure supported on a line. The main features of the method are as: i) its computational complexity is on par with SW, ii) it provides an optimal transport plan through the 1D assignment problem, iii) it acts as an upper bound of WD, and iv) is amenable to optimization to find the optimal pivot measure. As an additional contribution, we establish a closed form of the WD when an input measure is supported on a line.

**Outline.** Section 2 presents some background of OT. Section 3 formulates our new WD proxy, provides some of its topological properties and a numerical computation scheme. Section 4 builds upon the concept of Wasserstein generalized geodesics to reformulate our OT metric approximation as the Sliced Wasserstein Generalized Geodesics (SWGG) along its optimal variant coined min-SWGG. This reformulation allows deriving additional topological properties and an optimization scheme. Finally, Section 5 provides experimental evaluations.

**Notations.** Let $\langle \cdot, \cdot \rangle$ be the Euclidean inner product on $\mathbb{R}^d$ and let $\mathbb{S}^{d-1} = \{ \boldsymbol{u} \in \mathbb{R}^d \text{ s.t. } \|\boldsymbol{u}\|_2 = 1 \}$, the unit sphere. We denote $\mathcal{P}(\mathbb{R}^d)$ the set of probability measures on $\mathbb{R}^d$ endowed with the $\sigma-$algebra of Borel set and $\mathcal{P}_2(\mathbb{R}^d) \subset \mathcal{P}(\mathbb{R}^d)$ those with finite second-order moment i.e. $\mathcal{P}_2(\mathbb{R}^d) = \{ \mu \in \mathcal{P}(\mathbb{R}^d) \text{ s.t. } \int_{\mathbb{R}^d} \|\boldsymbol{x}\|_2^2 d\mu(\boldsymbol{x}) < \infty \}$. Let $\mathcal{P}_2^n(\mathbb{R}^d)$ be the subspace of $\mathcal{P}_2(\mathbb{R}^d)$ defined by empirical measures with $n$-atoms and uniform masses. For any measurable function $f : \mathbb{R}^d \to \mathbb{R}^d$, we denote $f_\#$ its push forward, namely for $\mu \in \mathcal{P}_2(\mathbb{R}^d)$ and for any measurable set $A \in \mathbb{R}^d$, $f_\# \mu(A) = \mu(f^{-1}(A))$, with $f^{-1}(A) = \{ \boldsymbol{x} \in \mathbb{R}^d \text{ s.t. } f(\boldsymbol{x}) \in A \}$.

## 2 Background on Optimal Transport

**Definition 2.1** (Wasserstein distance)**.** The squared WD [63] between $\mu_1, \mu_2 \in \mathcal{P}_2(\mathbb{R}^d)$ is defined as:

$$W_2^2(\mu_1, \mu_2) \overset{\text{def}}{=} \inf_{\pi \in \Pi(\mu_1, \mu_2)} \int_{\mathbb{R}^d \times \mathbb{R}^d} \|\boldsymbol{x} - \boldsymbol{y}\|_2^2 d\pi(\boldsymbol{x}, \boldsymbol{y}) \tag{1}$$

with $\Pi(\mu_1, \mu_2) = \{\pi \in \mathcal{P}_2(\mathbb{R}^d \times \mathbb{R}^d)$ s.t. $\pi(A \times \mathbb{R}^d) = \mu_1(A)$ and $\pi(\mathbb{R}^d \times A) = \mu_2(A), \forall A$ measurable set of $\mathbb{R}^d\}$.

The $\arg\min$ of Eq. (1) is referred to as the optimal transport plan. Denoted $\pi^*$, it expresses how to move the probability mass from $\mu_1$ to $\mu_2$ with minimum cost. In some cases, $\pi^*$ is of the form $(Id, T)_{\#}\mu_1$ for a measurable map $T : \mathbb{R}^d \to \mathbb{R}^d$, *i.e.* there is no mass splitting during the transport. This map is called a Monge map and is denoted $T^{\mu_1 \to \mu_2}$ (or shortly $T^{1\to2}$). Thus, one has $W_2^2(\mu_1, \mu_2) = \inf_{T \text{ s.t. } T_{\#}\mu_1=\mu_2} \int_{\mathbb{R}^d} \|\boldsymbol{x} - T(\boldsymbol{x})\|_2^2 d\mu_1(\boldsymbol{x})$. This occurs, for instance, when $\mu_1$ has a density w.r.t. the Lebesgue measure [12] or when $\mu_1$ and $\mu_2$ are in $\mathcal{P}_2^n(\mathbb{R}^d)$ [58].

Endowed with the WD, the space $\mathcal{P}_2(\mathbb{R}^d)$ is a geodesic space. Indeed, since there exists a Monge map $T^{1\to2}$ between $\mu_1$ and $\mu_2$, one can define a geodesic curve $\mu^{1\to2} : [0,1] \to \mathcal{P}_2(\mathbb{R}^d)$ [31] as:

$$\forall t \in [0,1], \ \mu^{1\to2}(t) \stackrel{\text{def}}{=} (tT^{1\to2} + (1-t)Id)_{\#}\mu_1 \tag{2}$$

which represents the shortest path w.r.t. Wasserstein distance in $\mathcal{P}_2(\mathbb{R}^d)$ between $\mu_1$ and $\mu_2$. The Wasserstein mean between $\mu_1$ and $\mu_2$ corresponds to $t = 0.5$ and we simply write $\mu^{1\to2}$.

This notion of geodesic allows the study of the curvature of the Wasserstein space [1]. Indeed, the Wasserstein space is of positive curvature [51], *i.e.* it respects the following inequality:

$$W_2^2(\mu_1, \mu_2) \geq 2W_2^2(\mu_1, \nu) + 2W_2^2(\nu, \mu_2) - 4W_2^2(\mu^{1\to2}, \nu) \tag{3}$$

for all pivot measures $\nu \in \mathcal{P}_2(\mathbb{R}^d)$.

**Solving and approximating Optimal Transport.** The Wasserstein distance between empirical measures $\mu_1, \mu_2$ with $n$-atoms can be computed in $\mathcal{O}(n^3 \log n)$, preventing from the use of OT for large scale applications [11]. Several algorithms have been proposed to lower this complexity, for example the Sinkhorn algorithm [23] that provides an approximation in near $\mathcal{O}(n^2)$ complexity [2].

Notably, when $\mu_1 = \frac{1}{n}\sum_{i=1}^n \delta_{x_i}$ and $\mu_2 = \frac{1}{n}\sum_{i=1}^n \delta_{y_i}$ are 1D distributions, computing the WD can be done by matching the sorted empirical samples, leading to an overall complexity of $\mathcal{O}(n \log n)$. More precisely, let $\sigma$ and $\tau$ two permutation operators s.t. $x_{\sigma(1)} \leq x_{\sigma(2)} \leq ... \leq x_{\sigma(n)}$ and $y_{\tau(1)} \leq y_{\tau(2)} \leq ... \leq y_{\tau(n)}$. Then, the 1D Wasserstein distance is given by:

$$W_2^2(\mu_1, \mu_2) = \frac{1}{n}\sum_{i=1}^n (x_{\sigma(i)} - y_{\tau(i)})^2. \tag{4}$$

**Sliced WD.** The Sliced-Wasserstein distance (SWD) [56] aims to scale up the computation of OT by leveraging the closed form expression (4) of the Wasserstein distance for 1D distributions. It is defined as the expectation of 1D-WD computed along projection directions $\theta \in \mathbb{S}^{d-1}$ over the unit sphere:

$$\text{SW}_2^2(\mu_1, \mu_2) \stackrel{\text{def}}{=} \int_{\mathbb{S}^{d-1}} W_2^2(P_{\#}^\theta \mu_1, P_{\#}^\theta \mu_2) d\omega(\theta), \tag{5}$$

where $P_{\#}^\theta \mu_1$ and $P_{\#}^\theta \mu_2$ are projections onto the direction $\theta \in \mathbb{S}^{d-1}$ with $P^\theta : \mathbb{R}^d \to \mathbb{R}, \boldsymbol{x} \mapsto \langle \boldsymbol{x}, \theta \rangle$ and where $\omega$ is the uniform distribution over $\mathbb{S}^{d-1}$.

Since the integral in Eq. (5) is intractable, one resorts, in practice, to Monte-Carlo estimation to approximate the SWD.

Its computation only involves projections and permutations. For $L$ directions, the computational complexity is $\mathcal{O}(dLn + Ln\log n)$ and the memory complexity is $\mathcal{O}(Ld + Ln)$. However, in high dimension, several projections are necessary to approximate accurately the SWD and many projections lead to 1D-WD close to 0. This issue is well known in the SW community [68], where different ways of performing effective sampling have been proposed [49, 46, 50] such as distributional or hierarchical slicing. In particular, this motivates the definition of max-Sliced-Wasserstein [24] which keeps only the most informative slice:

$$\text{max-SW}_2^2(\mu_1, \mu_2) \stackrel{\text{def}}{=} \max_{\theta \in \mathbb{S}^{d-1}} W_2^2(P_{\#}^\theta \mu_1, P_{\#}^\theta \mu_2). \tag{6}$$

While being a non convex problem, it can be optimized efficiently using a gradient ascent scheme.

The SW-like distances are attractive since they are fast to compute and enjoy theoretical properties: they are proper metrics and metricize the weak convergence. However, they do not provide an OT plan.

**Projected WD.** Another quantity of interest based on the 1D-WD is the projected Wasserstein distance (PWD) [57]. It leverages the permutations of the projected distributions in 1D in order to derive couplings between the original distributions.

Let $\mu_1 = \frac{1}{n} \sum_{i=1}^n \delta_{\boldsymbol{x}_i}$ and $\mu_2 = \frac{1}{n} \sum_{i=1}^n \delta_{\boldsymbol{y}_i}$ in $\mathcal{P}_2^n(\mathbb{R}^d)$. The PWD is defined as:

$$\mathrm{PWD}_2^2(\mu_1, \mu_2) \stackrel{\text{def}}{=} \int_{\mathbb{S}^{d-1}} \frac{1}{n} \sum_{i=1}^n \|\boldsymbol{x}_{\sigma_\theta(i)} - \boldsymbol{y}_{\tau_\theta(i)}\|_2^2 d\omega(\theta), \tag{7}$$

where $\sigma_\theta, \tau_\theta$ are the permutations obtained by sorting $P_\#^\theta \mu_1$ and $P_\#^\theta \mu_2$.

As some permutations are not optimal, we straightforwardly have $W_2^2 \leq \mathrm{PWD}_2^2$. Note that some permutations can appear highly irrelevant in the original space, leading to an overestimation of $W_2^2$ (typically when the distributions are multi-modal or with support lying in a low dimensional manifold, see Supp. 7.1 for a discussion).

In this paper, we restrict ourselves to empirical distributions with the same number of samples. They are defined as $\mu_1 = \frac{1}{n} \sum_{i=1}^n \delta_{\boldsymbol{x}_i}$ and $\mu_2 = \frac{1}{n} \sum_{i=1}^n \delta_{\boldsymbol{y}_i}$ in $\mathcal{P}_2^n(\mathbb{R}^d)$. Note that the results presented therein can be extended to any discrete measures by mainly using quantile functions instead of permutations and transport plans instead of transport maps (see Supp. 7.2).

## 3 Definition and properties of min-SWGG

The fact that the PWD overestimates $W_2^2$ motivates the introduction of our new loss function coined min-SWGG which keeps only the most informative permutation. Afterwards, we derive a property of distance and grant an estimation of min-SWGG via random search of the directions.

**Definition 3.1** (SWGG and min-SWGG). Let $\mu_1, \mu_2 \in \mathcal{P}_2^n(\mathbb{R}^d)$ and $\theta \in \mathbb{S}^{d-1}$. Denote by $\sigma_\theta$ and $\tau_\theta$ the permutations obtained by sorting the 1D projections $P_\#^\theta \mu_1$ and $P_\#^\theta \mu_2$. We define respectively SWGG and min-SWGG as:

$$\mathrm{SWGG}_2^2(\mu_1, \mu_2, \theta) \stackrel{\text{def}}{=} \frac{1}{n} \sum_{i=1}^n \|\boldsymbol{x}_{\sigma_\theta(i)} - \boldsymbol{y}_{\tau_\theta(i)}\|_2^2, \tag{8}$$

$$\text{min-SWGG}_2^2(\mu_1, \mu_2) \stackrel{\text{def}}{=} \min_{\theta \in \mathbb{S}^{d-1}} \mathrm{SWGG}_2^2(\mu_1, \mu_2, \theta). \tag{9}$$

One shall remark that the function SWGG corresponds to the building block of PWD in eq. (7).

One main feature of min-SWGG is that it comes with a transport map. Let $\theta^* \in$ argmin $\mathrm{SWGG}_2^2(\mu_1, \mu_2, \theta)$ be the optimal projection direction. The associated transport map is:

$$T(\boldsymbol{x}_i) = \boldsymbol{y}_{\tau_{\theta^*}^{-1}(\sigma_{\theta^*}(i))}, \quad \forall 1 \leq i \leq n. \tag{10}$$

In Supp. 7.6 we give several examples of such transport plan. These examples show that the overall structure of the optimal transport plan is respected by the transport plan obtained via min-SWGG.

We now give some theoretical properties of the quantities min-SWGG and SWGG. Their proofs are given in Supp. 7.3.

**Proposition 3.2** (Distance and Upper bound). Let $\theta \in \mathbb{S}^{d-1}$. $\mathrm{SWGG}_2(\cdot, \cdot, \theta)$ defines a distance on $\mathcal{P}_2^n(\mathbb{R}^d)$. Moreover, min-SWGG is an upper bound of $W_2^2$, and $W_2^2 \leq \text{min-SWGG}_2^2 \leq \mathrm{PWD}_2^2$, with equality between $W_2^2$ and min-SWGG$_2^2$ when $d > 2n$.

**Remark 3.3.** Similarly to max-SW, min-SWGG retains only one optimal direction $\theta^* \in \mathbb{S}^{d-1}$. However, the two distances strongly differ: i) min-SWGG is an upper bound and max-SW a lower bound of $W_2^2$, ii) the optimal $\theta^*$ may differ (see Supp. 7.4 for an illustration), and iii) max-SW does not provide a transport plan between $\mu_1$ and $\mu_2$.

Solving Eq. (9) can be achieved using a random search, by sampling $L$ directions $\theta \in \mathbb{S}^{d-1}$ and keeping only the one leading to the lowest value of SWGG.

This gives an overall computational complexity of $\mathcal{O}(Ldn + Ln \log n)$ and a memory complexity of $\mathcal{O}(dn)$. In low dimension, the random search estimation is effective: covering all possible

permutations through $\mathbb{S}^{d-1}$ can be done with a low number of directions. In high dimension, many more directions $\theta$ are needed to have a relevant approximation, typically $\mathcal{O}(L^{d-1})$. This motivates the design of gradient descent techniques for finding $\theta^*$.

# 4   SWGG as minimizing along the Wasserstein generalized geodesics

Solving problem in Eq. (9) amounts to optimize over a set of admissible permutations. This problem is hard since SWGG is non convex w.r.t. $\theta$ and piecewise constant, thus not differentiable over $\mathbb{S}^{d-1}$. Indeed, as long as the permutations remain the same for different directions $\theta$, the value of SWGG remains constant. When the permutations change, the objective SWGG "jumps" as illustrated in Fig. 1.

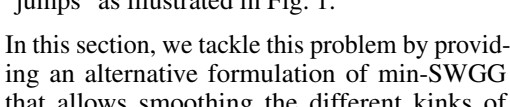 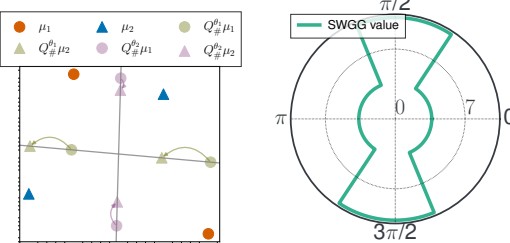

Figure 1: (Left) Empirical distributions with examples of 2 sampled lines (Right) that lead to 2 possible values of SWGG when $\theta \in [0, 2\pi]$.

In this section, we tackle this problem by providing an alternative formulation of min-SWGG that allows smoothing the different kinks of SWGG, hence, making min-SWGG amenable to optimization. This formulation relies on Wasserstein generalized geodesics we introduce hereinafter.

We show that this alternative formulation brings in computational advantages and allows establishing some additional topological properties and deriving an efficient optimization scheme. We also provide a new closed form expression of the Wasserstein distance $W_2^2(\mu_1, \mu_2)$ when either $\mu_1$ or $\mu_2$ is supported on a line.

## 4.1   SWGG based on Wasserstein Generalized Geodesics

Wasserstein generalized geodesics (see Supp. 8 for more details) were first introduced in [4] in order to ensure the convergence of Euler scheme for Wasserstein Gradient Flows. This concept has been used notably in [29, 44] to speed up some computations and to derive some theoretical properties. Generalized geodesic is also highly related with the idea of linearization of the Wasserstein distance via an $L^2$ space [65, 43], see Supp. 9 for more details on the related works.

Generalized geodesics lay down on a pivot measure $\nu \in \mathcal{P}_2^n(\mathbb{R}^d)$ to transport the distribution $\mu_1$ toward $\mu_2$. Indeed, one can leverage the optimal transport maps $T^{\nu \to \mu_1}$ and $T^{\nu \to \mu_2}$ to construct a curve $t \mapsto \mu_g^{1 \to 2}(t)$ linking $\mu_1$ to $\mu_2$ as

$$\mu_g^{1 \to 2}(t) \overset{\text{def}}{=} ((1-t)T^{\nu \to \mu_1} + tT^{\nu \to \mu_2})_{\#}\, \nu, \qquad \forall t \in [0, 1]. \tag{11}$$

The related generalized Wasserstein mean corresponds to $t = 0.5$ and is denoted $\mu_g^{1 \to 2}$.

Intuitively, the optimal transport maps between $\nu$ and $\mu_i, i = 1, 2$ give rise to a sub-optimal transport map between $\mu_1$ and $\mu_2$:

$$T_{\nu}^{1 \to 2} \overset{\text{def}}{=} T^{\nu \to \mu_2} \circ T^{\mu_1 \to \nu} \quad \text{with} \quad (T_{\nu}^{1 \to 2})_{\#}\mu_1 = \mu_2. \tag{12}$$

One can be interested in the cost induced by the transportation of $\mu_1$ to $\mu_2$ via the transport map $T_{\nu}^{1 \to 2}$, known as the $\nu$-based Wasserstein distance [47] and defined as

$$W_{\nu}^2(\mu_1, \mu_2) \overset{\text{def}}{=} \int_{\mathbb{R}^d} \|\boldsymbol{x} - T_{\nu}^{1 \to 2}(\boldsymbol{x})\|_2^2 d\mu_1(\boldsymbol{x}) = 2W_2^2(\mu_1, \nu) + 2W_2^2(\nu, \mu_2) - 4W_2^2(\mu_g^{1 \to 2}, \nu). \tag{13}$$

Notably, the second part of Eq. (13) straddles the square Wasserstein distance with Eq. (3). Remarkably, the computation of $W_{\nu}^2$ can be efficient if the pivot measure $\nu$ is chosen appropriately. As established in Lemma 4.6, it is the case when $\nu$ is supported on a line. Based on these facts, we propose hereafter an alternative formulation of SWGG.

**Definition 4.1** (Pivot measure). Let $\mu_1$ and $\mu_2 \in \mathcal{P}_2^n(\mathbb{R}^d)$. We restrict the pivot measure $\nu$ to be the Wasserstein mean of the measures $Q_{\#}^{\theta}\mu_1$ and $Q_{\#}^{\theta}\mu_2$:

$$\mu_{\theta}^{1 \to 2} \overset{\text{def}}{=} \arg \min_{\mu \in \mathcal{P}_2^n(\mathbb{R}^d)} W_2^2(Q_{\#}^{\theta}\mu_1, \mu) + W_2^2(\mu, Q_{\#}^{\theta}\mu_2),$$

where $\theta \in \mathbb{S}^{d-1}$ and $Q^\theta : \mathbb{R}^d \to \mathbb{R}^d$, $\boldsymbol{x} \mapsto \theta\langle \boldsymbol{x}, \theta\rangle$ is the projection onto the subspace generated by $\theta$. Moreover $\mu_\theta^{1\to 2}$ is always defined as the middle of a geodesic as in Eq (2).

One shall notice that $Q_{\#}^\theta \mu_1$ and $Q_{\#}^\theta \mu_2$ are supported on the line defined by the direction $\theta$, so is the pivot measure $\nu = \mu_\theta^{1\to 2}$. We are now ready to reformulate the metric SWGG.

**Proposition 4.2** (SWGG based on generalized geodesics). Let $\theta \in \mathbb{S}^{d-1}$, $\mu_1, \mu_2 \in \mathcal{P}_2^n(\mathbb{R}^d)$ and $\mu_\theta^{1\to 2}$ be the pivot measure. Let $\mu_{g,\theta}^{1\to 2}$ be the generalized Wasserstein mean between $\mu_1$ and $\mu_2 \in \mathcal{P}_2^n(\mathbb{R}^d)$ with pivot measure $\mu_\theta^{1\to 2}$. Then,

$$\mathrm{SWGG}_2^2(\mu_1, \mu_2, \theta) = 2W_2^2(\mu_1, \mu_\theta^{1\to 2}) + 2W_2^2(\mu_\theta^{1\to 2}, \mu_2) - 4W_2^2(\mu_{g,\theta}^{1\to 2}, \mu_\theta^{1\to 2}). \tag{14}$$

The proof is in Supp.10.1. From Proposition 4.2, SWGG is the $\mu_\theta^{1\to 2}$-based Wasserstein distance between $\mu_1$ and $\mu_2$. This alternative formulation allows establishing additional properties of min-SWGG.

## 4.2 Theoretical properties

Additionally to the properties derived in Section 3 (SWGG is a distance and min-SWGG is an upper bound of $W_2^2$), we provide below other theoretical guarantees.

**Proposition 4.3** (Weak Convergence). min-SWGG metricizes the weak convergence in $\mathcal{P}_2^n(\mathbb{R}^d)$. In other words, let $(\mu_k)_{k\in\mathbb{N}}$ be a sequence of measures in $\mathcal{P}_2^n(\mathbb{R}^d)$ and $\mu \in \mathcal{P}_2^n(\mathbb{R}^d)$. We have:

$$\mu_k \xrightarrow[k]{\mathcal{L},2} \mu \iff \text{min-SWGG}_2^2(\mu_k, \mu) \xrightarrow[k]{} 0,$$

where $\xrightarrow{\mathcal{L},2}$ stands for the weak convergence of measure i.e. $\int_{\mathbb{R}^d} f d\mu_k \to \int_{\mathbb{R}^d} f d\mu$ for all continuous bounded functions $f$.

Beyond the weak convergence, min-SWGG possesses the translation property, *i.e.* the translations can be factored out as the Wasserstein distance does (see [53, remark 2.19] for a recall).

**Proposition 4.4** (Translation). Let $T^u$ (resp. $T^v$) be the map $\boldsymbol{x} \mapsto \boldsymbol{x} - \boldsymbol{u}$ (resp. $\boldsymbol{x} \mapsto \boldsymbol{x} - \boldsymbol{v}$), with $\boldsymbol{u}, \boldsymbol{v}$ vectors of $\mathbb{R}^d$. We have:

$$\text{min-SWGG}_2^2(T_{\#}^u\mu_1, T_{\#}^v\mu_2) = \text{min-SWGG}_2^2(\mu_1, \mu_2) + \|\boldsymbol{u} - \boldsymbol{v}\|_2^2 - 2\langle \boldsymbol{u} - \boldsymbol{v}, \boldsymbol{m_1} - \boldsymbol{m_2}\rangle$$

where $\boldsymbol{m_1} = \int_{\mathbb{R}^d} \boldsymbol{x} d\mu_1(\boldsymbol{x})$ and $\boldsymbol{m_2} = \int_{\mathbb{R}^d} \boldsymbol{x} d\mu_2(\boldsymbol{x})$ are the means of $\mu_1, \mu_2$.

This property is useful in some applications such as shape matching, in which translation invariances are sought.

The proofs of the two Propositions are deferred to Supp. 10.2 and 10.3.

**Remark 4.5** (Equality). min-SWGG and $W_2^2$ are equal in different cases. First, [43] showed that it is the case whenever $\mu_1$ is the shift and scaling of $\mu_2$ (see Supp. 9.1 for a full discussion). In Lemma 4.6, we will state that it is also the case if one of the two distributions is supported on a line.

## 4.3 Efficient computation of SWGG

SWGG defined in Eq. (14) involves computing three WDs that are fast to compute, with an overall $\mathcal{O}(dn + n \log n)$ complexity, as detailed below. Building on this result, we provide an optimization scheme that allows optimizing over $\theta$ with $\mathcal{O}(sdn + sn \log sn)$ operations at each iteration, with $s$ a (small) integer. We first start by giving a new closed form expression of the WD whenever one distribution is supported on a line, that proves useful for deriving an efficient computation scheme.

**New closed form of the WD.** The following lemma states that $W_2^2(\mu_1, \mu_2)$ admits a closed form whenever $\mu_2$ is supported on a line.

This lemma leverages the computation of the WD between $\mu_2$ and the orthogonal projection of $\mu_1$ onto the linear subspace defined by the line. Additionally, it provides an explicit formulation for the optimal transport map $T^{1\to 2}$.

**Lemma 4.6.** Let $\mu_1, \mu_2$ in $\mathcal{P}_2^n(\mathbb{R}^d)$ with $\mu_2$ supported on a line of direction $\theta \in \mathbb{S}^{d-1}$. We have:

$$W_2^2(\mu_1, \mu_2) = W_2^2(\mu_1, Q_\#^\theta \mu_1) + W_2^2(Q_\#^\theta \mu_1, \mu_2) \tag{15}$$

with $Q^\theta$ as in Def. 4.1. Note that $W_2^2(\mu_1, Q_\#^\theta \mu_1) = \frac{1}{n} \sum \|\boldsymbol{x}_i - Q^\theta(\boldsymbol{x}_i)\|_2^2$ and $W_2^2(Q_\#^\theta \mu_1, \mu_2) = W_2^2(P_\#^\theta \mu_1, P_\#^\theta \mu_2)$ are the WD between 1D distributions. Additionally, the optimal transport map is given by $T^{1 \to 2} = T^{Q_\#^\theta \mu_1 \to \mu_2} \circ T^{\mu_1 \to Q_\#^\theta \mu_1} = T^{Q_\#^\theta \mu_1 \to \mu_2} \circ Q^\theta$. In particular, the map $T^{1 \to 2}$ can be obtained via the permutations of the 1D distributions $P_\#^\theta \mu_1$ and $P_\#^\theta \mu_2$. The proof is provided in Supp. 10.4.

**Efficient computation of** SWGG.    Eq. (14) is defined as the Wasserstein distance between a distribution (either $\mu_1$ or $\mu_2$ or $\mu_{g,\theta}^{1 \to 2}$) and a distribution supported on a line ($\mu_\theta^{1 \to 2}$). As detailed in Supp. 10.5, computation of Eq. (14) involves three Wasserstein distances between distributions and their projections: i) $W_2^2(\mu_1, Q_\#^\theta \mu_1)$, ii) $W_2^2(\mu_2, Q_\#^\theta \mu_2)$, iii) $W_2^2(\mu_{g,\theta}^{1 \to 2}, \mu_\theta^{1 \to 2})$, and a one dimensional Wasserstein distance $W_2^2(P_\#^\theta \mu_1, P_\#^\theta \mu_2)$, resulting in a $\mathcal{O}(dn + n \log n)$ complexity.

**Optimization scheme for min-SWGG.**    The term $W_2^2(\mu_{g,\theta}^{1 \to 2}, \mu_\theta^{1 \to 2})$ in Eq. (14) is not continuous w.r.t. $\theta$. Indeed, the generalized mean $\mu_{g,\theta}^{1 \to 2}$ depends only on the transport maps $T^{\mu_\theta^{1 \to 2} \to \mu_1}$ and $T^{\mu_\theta^{1 \to 2} \to \mu_2}$, which remain constant as long as different projection directions $\theta$ lead to the same permutations $\sigma_\theta$ and $\tau_\theta$. Hence, we rely on a smooth surrogate $\widetilde{\mu_{g,\theta}^{1 \to 2}}$ of the generalized mean and we aim to minimize the following objective function:

$$\widetilde{\text{SWGG}}_2^2(\mu_1, \mu_2, \theta) \overset{\text{def}}{=} 2W_2^2(\mu_1, \mu_\theta^{1 \to 2}) + 2W_2^2(\mu_\theta^{1 \to 2}, \mu_2) - 4W_2^2(\widetilde{\mu_{g,\theta}^{1 \to 2}}, \mu_\theta^{1 \to 2}). \tag{16}$$

To define $\widetilde{\mu_{g,\theta}^{1 \to 2}}$, one option would be to use entropic maps in Eq. (11) but at the price of a quadratic time complexity. We rather build upon the blurred Wasserstein distance [26] to define $\widetilde{\mu_{g,\theta}^{1 \to 2}}$ as it can be seen as an efficient surrogate of entropic transport plans in 1D. In one dimensional setting, $\widetilde{\mu_{g,\theta}^{1 \to 2}}$ can be approximated efficiently by adding an empirical Gaussian noise followed by a sorting pass. In our case, it resorts in making $s$ copies of each sorted projection $P^\theta(\boldsymbol{x}_{\sigma(i)})$ and $P^\theta(\boldsymbol{y}_{\tau(i)})$ respectively, to add an empirical Gaussian noise of deviation $\sqrt{\epsilon}/2$ and to compute averages of sorted blurred copies $\boldsymbol{x}_{\sigma^s}^s, \boldsymbol{y}_{\tau^s}^s$. We finally have $(\widetilde{\mu_{g,\theta}^{1 \to 2}})_i = \frac{1}{2s} \sum_{k=(i-1)s+1}^{is} \boldsymbol{x}_{\sigma^s(k)}^s + \boldsymbol{y}_{\tau^s(k)}^s$. [26] showed that this blurred WD has the same asymptotic properties as the Sinkhorn divergence.

The surrogate $\widetilde{\text{SWGG}}(\mu_1, \mu_2, \theta)$ is smoother w.r.t. $\theta$ and can thus be optimized using gradient descent, converging towards a local minima. Once the optimal direction $\theta^*$ is found, min-SWGG resorts to be the solution provided by $\text{SWGG}(\mu_1, \mu_2, \theta^*)$. Fig. 2 illustrates the effect of the smoothing on a toy example and more details are given in Supp. 10.6. The computation of $\widetilde{\text{SWGG}}(\mu_1, \mu_2, \theta)$ is summarized in Alg. 1.

---

**Algorithm 1** Computing $\widetilde{\text{SWGG}}_2^2(\mu_1, \mu_2, \theta)$

---

**Require:** $\mu_1 = \frac{1}{n} \sum \delta_{\boldsymbol{x}_i}, \mu_2 = \frac{1}{n} \sum \delta_{\boldsymbol{y}_i}, \theta \in \mathbb{S}^{d-1}, s \in \mathbb{N}_+$ and $\epsilon \in \mathbb{R}_+$

$\quad \sigma, \tau \leftarrow$ ascending ordering of $(P^\theta(\boldsymbol{x}_i))_i, (Q^\theta(\boldsymbol{y}_i))_i$

$\quad \boldsymbol{x}^s \leftarrow s$ copies of $(\boldsymbol{x}_{\sigma(i)})_i, \boldsymbol{y}^s \leftarrow s$ copies of $(\boldsymbol{y}_{\tau(i)})_i$

$\quad \sigma^s, \tau^s \leftarrow$ ascending ordering of $\langle \boldsymbol{x}^s, \theta \rangle + \boldsymbol{\xi}, \langle \boldsymbol{y}^s, \theta \rangle + \boldsymbol{\xi}$ for $\xi_i \sim \mathcal{N}(0, \epsilon/2), \forall i \leq sn$

$\quad a \leftarrow \frac{2}{n} \sum_i \left( \|\boldsymbol{x}_i - Q^\theta(\boldsymbol{x}_i)\|_2^2 + \|\boldsymbol{y}_i - Q^\theta(\boldsymbol{y}_i)\|_2^2 \right) \qquad \triangleright 2W_2^2(\mu_1, Q_\#^\theta \mu_1) + 2W_2^2(\mu_2, Q_\#^\theta \mu_2)$

$\quad b \leftarrow \frac{2}{n} \sum_i \|P^\theta(\boldsymbol{x}_{\sigma(i)}) + P^\theta(\boldsymbol{x}_{\tau(i)})\|_2^2 \qquad \triangleright 2W_2^2(P_\#^\theta \mu_1, P_\#^\theta \mu_2)$

$\quad c \leftarrow \frac{4}{n} \sum_i \left\| \frac{1}{2}(Q^\theta(\boldsymbol{x}_{\sigma(i)}) + Q^\theta(\boldsymbol{y}_{\tau(i)})) - \frac{1}{2s} \sum_{k=(i-1)s+1}^{is} (\boldsymbol{x}_{\sigma^s(k)}^s + \boldsymbol{y}_{\tau^s(k)}^s) \right\|_2^2 \triangleright 4W_2^2(\widetilde{\mu_{g,\theta}^{1 \to 2}}, \mu_\theta^{1 \to 2})$

$\quad$ **Output** $a + b - c$

---

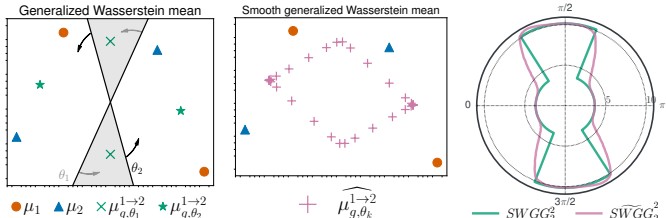

Figure 2: Illustration of the smoothing effect in the same setting as in Fig. 1. (Left) Two sets of generalized Wasserstein means are possible, depending on the direction of the sampled line w.r.t. $\theta_1$ and $\theta_2$, giving rise to 2 different values for SWGG. (Middle) The surrogate provides a smooth transition between the two sets of generalized Wasserstein means as the direction $\theta$ changes, (Right) providing a smooth approximation of SWGG that is amenable to optimization.

## 5   Experiments

We highlight that min-SWGG is fast to compute, gives an approximation of the WD and the associated transport plan. We start by comparing the random search and the gradient descent schemes for finding the optimal direction in subsection 5.1. Subsection 5.2 illustrates the weak convergence property of min-SWGG through a gradient flow application to match distributions. We then implement an efficient algorithm for colorization of gray scale images in 5.3, thanks to the new closed form expression of the WD. We finally evaluate min-SWGG in a shape matching context in subsection 5.4. When possible from the context, we compare min-SWGG with the main methods for approximating the WD namely SW, max-SW, Sinkhorn [23], factored coupling [29] and subspace robust WD (SRW) [52]. Supp. 11 provides additional results on the behavior of min-SWGG and experiments on other tasks such as color transfer or on data sets distance computation. All the code is available at [1]

### 5.1   Computing min-SWGG

Let consider Gaussian distributions in dimensions $d \in \{2, 20, 200\}$. We first sample $n = 1000$ points from each distribution to define $\mu_1$ and $\mu_2$. We then compute min-SWGG$_2^2(\mu_1, \mu_2)$ computed using different schemes, either by random search, by simulated annealing [54] or by gradient descent. We report the obtained results in Fig. 3 (left). For the random search scheme, we repeat each experiment 20 times and we plot the average value of min-SWGG $\pm 2$ times the standard deviation.

For the gradient descent, we select a random initial $\theta$. We observe that, in low dimension, all schemes provide similar values of min-SWGG. When the dimension increases, optimizing the direction $\theta$ yields a more accurate approximation of the true Wasserstein distance (see plots' title in Fig. 3). On Fig. 3 (right), we compare the empirical runtime evaluation for min-SWGG with different competitors for $d = 3$ and using $n$ samples from Gaussian distributions, with $n \in \{10^2, 10^3, 10^4, 5 \times 10^4, 10^5\}$. We observe that, as expected, min-SWGG with random search is as fast as SW with a super linear time complexity. With the optimization process, it is faster than SRW for a given number of samples. We also note that SRW is more demanding in memory and hence does not scale as well as min-SWGG. We give more details on this experimentation and a comparison with competitors in Supp. 11.2.

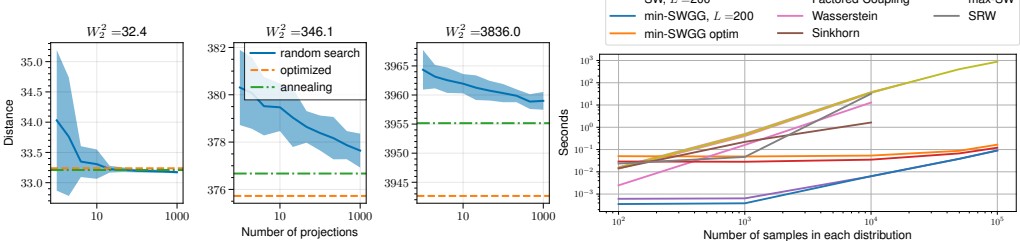

Figure 3: (Left) evolution of min-SWGG with different numbers of projections and with the dimension $d$ in $\{2, 20, 200\}$. (Right) Runtimes.

---

[1] https://github.com/MaheyG/SWGG

## 5.2 Gradient Flows

We highlight the weak convergence property of min-SWGG. Initiating from a random initial distribution, we aim to move the particles of a source distribution $\mu_1$ towards a target one $\mu_2$ by reducing the objective min-SWGG$_2^2(\mu_1, \mu_2)$ at each step. We compare both variants of min-SWGG against SW, max-SW and PWD, relying on the code provided in [37] for running the experiment; we report the results on Fig. 4. We consider several target distributions, representing diverse scenarios and fix $n = 100$. We run each experiment 10 times and report the mean $\pm$ the standard deviation. In every case, one can see that $\mu_1$ moves towards $\mu_2$ and that all methods tend to have similar behavior. One can notice though that, for the distributions in $d = 500$ dimensional space, min-SWGG computed with the optimization scheme leads to the best alignment of the distributions.

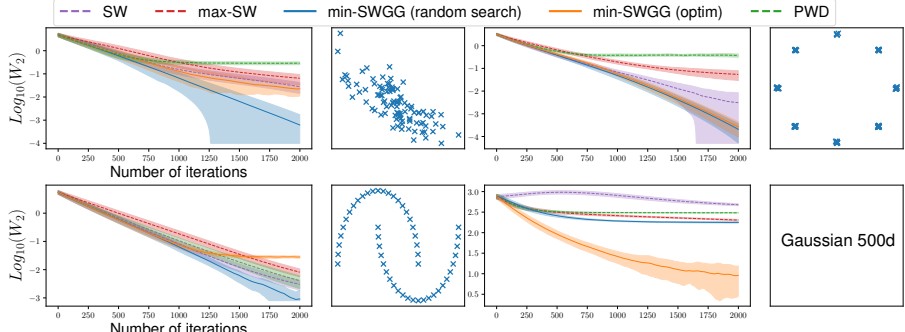

Figure 4: Log of the WD between different source and target distributions as a function of the number of iterations.

## 5.3 Gray scale image colorization

Lemma 4.6 states that the WD has a closed form when one of the 2 distributions is supported on a line, allowing us to compute the WD and the OT map with a complexity of $\mathcal{O}(dn + n \log n)$. This particular situation arises for instance with RBG images ($\mu_1, \mu_2 \in \mathcal{P}_2^n(\mathbb{R}^3)$), where black and white images are supported on a line (the line of grays). One can address the problem of image colorization through color transfer [25], where a black and white image is the source and a colorful image the target. Our fast procedure allows considering large images without sub-sampling with a reasonable computation time. Fig. 5 gives an example of colorization of an image of size $1280 \times 1024$ that was computed in less than 0.2 second, while being totally untractable for the $\mathcal{O}(n^3 \log n)$ solver of WD.

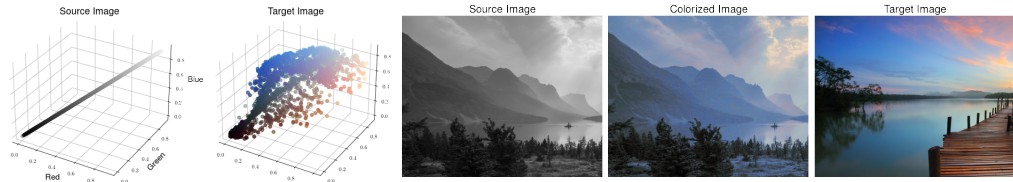

Figure 5: Cloud point source and target (left) colorization of image (right).

This procedure can be lifted to pan-sharpening [64] where one aims to construct a super-resolution multi-chromatic satellite image with the help of a super-resolution mono-chromatic image (source) and a low-resolution multi-chromatic image (target). Obtained results are given in the Supp. 11.4.

## 5.4 Point clouds registration

Iterative Closest Point (ICP) is an algorithm for aligning point clouds based on their geometries [7]. Roughly, its most popular version defines a one-to-one correspondence between point clouds, computes a rigid transformation (namely translation, rotation or reflection), moves the source point clouds using the transformation, and iterates the procedure until convergence. The rigid transformation is the solution of the Procrustes problem *i.e.* $\arg\min_{(\Omega, t) \in O(d) \times \mathbb{R}^d} \|\Omega(\boldsymbol{X} - t) - \boldsymbol{Y}\|_2^2$, where $\boldsymbol{X}, \boldsymbol{Y}$ are the source and the target cloud points and $O(d)$ the space of orthogonal matrices of dimension $d$. This Procrustes problem can be solved using a SVD [59] for instance.

We perform the ICP algorithm with different variants to compute the one-to-one correspondence: neareast neighbor (NN) correspondence, OT transport map (for small size datasets) and min-SWGG

transport map. Note that SW, PWD, SRW, factored coupling and Sinkhorn cannot be run in this context where a one-to-one correspondence is mandatory; subspace detours [44] are irrelevant in this context (see Supp. 11.5). We evaluate the results of the ICP algorithm in terms of: i) the quality of the final alignment, measured by the Sinkhorn divergence between the re-aligned and target point cloud; ii) the speed of the algorithm given by the running time until convergence. We consider 3 datasets of different sizes. The results are shown in Table 1 and more details about the setup, can be found in Supp. 11.5. In Supp. 11.5 we give a deeper analysis of the results, notably with different criteria for the final assignment, namely the Chamfer and the Frobenius distance. One can see that the assignment provided by OT-based methods is better than NN. min-SWGG allows working with large datasets, while OT fails to provide a solution for $n = 150000$.

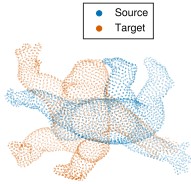

| $n$ | 500 | 3000 | 150 000 |
|---|---|---|---|
| NN | 3.54 (**0.02**) | 96.9 (**0.30**) | 23.3 (**59.37**) |
| OT | 0.32 (0.18) | 48.4 (58.46) | · |
| min-SWGG | **0.05** (0.04) | **37.6** (0.90) | **6.7** (105.75) |

Table 1: Sinkhorn Divergence between final transformation on the source and the target. Timings in seconds are into parenthesis. Best values are boldfaced. An example of a point clouds ($n = 3000$) is provided on the left.

## 6   Conclusion

In this paper, we hinge on the properties of sliced Wasserstein distance and on the Wasserstein generalized geodesics to define min-SWGG, a new upper bound of the Wasserstein distance that comes with an associated transport map. Topological properties of SWGG are provided, showing that it defines a metric and that min-SWGG metrizes the weak convergence of measure. We also propose two algorithms for computing min-SWGG, either through a random search scheme or a gradient descent procedure after smoothing the generalized geodesics definition of min-SWGG. We illustrate its behavior in several experimental setups, notably showcasing its interest in applications where a transport map is needed.

The set of permutations covered by min-SWGG is the one induced by projections and permutations on the line. It is a subset of the original Birkhoff polytope and it would be interesting to characterize how these two sets relates. In particular, in the case of empirical realizations of continuous distributions, the behavior of min-SWGG, when $n$ grows, needs to be investigated. In addition, the fact that min-SWGG and WD coincide when $d > 2n$ calls for embedding the distributions in higher dimensional spaces to benefit from the greater expressive power of projection onto the line. Another important consideration is to establish a theoretical upper bound for min-SWGG.

## Acknowledgments

The authors gratefully acknowledge the financial support of the French Agence Nationale de la Recherche (ANR), under grant ANR-20-CHIA-0021-01 (project RAIMO[2]), grant ANR-20-CHIA-0030 (project OTTOPIA) and grant ANR-18-CE23-0022-01 (project MULTISCALE). Clément Bonet is supported by the project DynaLearn from Labex CominLabs and Région Bretagne ARED DLearnMe.

---

[2]`https://chaire-raimo.github.io/`

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

# 7 Proofs and supplementary results related to Section 3

## 7.1 Overestimation of WD by PWD

As stated in Section 2, the projected Wasserstein distance PWD (see Eq. 7) tends to overestimate the Wasserstein distance. This is due to the fact that some permutations $\sigma_\theta$ and $\tau_\theta$ (with $\theta \in \mathbb{S}^{d-1}$) involved in PWD computation may be irrelevant. Such situation occurs when the distributions are in high dimension but supported on a low dimensional manifold or when the distributions are multi-modal.

Let consider the distributions $\mu_1$ and $\mu_2$ lying on a low dimensional manifold. In high dimension, randomly sampled vectors $\theta$ tend to be orthogonal. Moreover, vectors orthogonal to the low dimensional manifold lead to "collapsed" projected distributions $P_\#^\theta \mu_1$ and $P_\#^\theta \mu_2$ onto $\theta$. Hence, such projection directions lead to permutations that can be randomly chosen. To empirically illustrate this behavior of PWD, we consider $\mu_1$ and $\mu_2$ as Gaussian distributions in $\mathbb{R}^d$, $d = 10$ but supported on the first two coordinates and we sample 200 points per distribution. Table 2 summarizes the obtained corresponding distances and shows that PWD overestimates the WD.

Now, let us consider two multimodal distributions $\mu_1, \mu_2$ with $K$ clusters such that each cluster of $\mu_1$ has a close cluster from $\mu_2$ (cyclical monotonicity assumption). Also we assume the same number of points in each cluster. OT plan will match the corresponding clusters and will lead to a relatively low value for $W_2^2$ (since cluster from $\mu_1$ has a closely related cluster in $\mu_2$). However as PWD may allow permutations that make correspondences between points from different clusters (since a source cluster and a target cluster can be far in the original space but very close when projected on 1D), the resulting distance will be much more larger, leading to an overestimation of the Wasserstein distance. Table 2 provides an illustration for $K = 10$ clusters and $d = 2$.

Table 2: Values of $W_2^2$, PWD and min-SWGG on two toy examples. PWD samples $\theta$ uniformly over $\mathbb{S}^{d-1}$; PWD Orthogonal Projections seek orthogonal vectors (see [57] for more details)

| Distributions | Multi-modal | Low dimensional manifold |
|---|---|---|
| $W_2^2$ | 12 | 12 |
| PWD$_2^2$ Monte-Carlo | 54 | 29 |
| PWD$_2^2$ Orthogonal Projections | 54 | 37 |
| min-SWGG$_2^2$ | 13 | 13 |

## 7.2 Quantile version of SWGG

The main body of the paper expresses SWGG for empirical distributions $\mu_1$ and $\mu_2$ with the same number of points and uniform probability masses. In this section we derived SWGG in a more general setting of discrete distributions.

Let remark that min-SWGG relies on solving a 1D optimal transport (OT) problem. So far, the 1D OT problem was derived for $\mu_1, \mu_2 \in \mathcal{P}_2^n(\mathbb{R})$ and thus was expressed using the permutation operators $\tau$ and $\sigma$. In the general setting of distributions $\mu_1 \in \mathcal{P}_2^n(\mathbb{R})$ and $\mu_2 \in \mathcal{P}_2^m(\mathbb{R})$ with $n \neq m$, the 1D optimal transport is computed based on quantile functions. Hence, the expression of SWGG in the general setting of $\mu_1 \in \mathcal{P}_2^n(\mathbb{R})$ and $\mu_2 \in \mathcal{P}_2^m(\mathbb{R})$ hinges on quantile functions instead of permutations.

More formally, let $\mu \in \mathcal{P}_2^n(\mathbb{R})$; its cumulative function is defined as:

$$F_\mu : \mathbb{R} \to [0, 1] \ , \ x \mapsto \int_{-\infty}^{x} d\mu \tag{17}$$

and its quantile function (or pseudo inverse), is given by:

$$q_\mu : [0, 1] \to \mathbb{R} \ , \ r \mapsto \min\{x \in \mathbb{R} \cup \{-\infty\} \text{ s.t. } F_\mu(x) \geq r\} \tag{18}$$

An important remark is that the quantile function is a step function with $n$ (the number of atoms) discontinuities. Thus, it can be stored efficiently using two vectors of size $n$ (one for the locations of the discontinuities and the other for the values of the discontinuities).

For $\mu_1 \in \mathcal{P}_2^n(\mathbb{R})$ and $\mu_2 \in \mathcal{P}_2^m(\mathbb{R})$, we recover the Wasserstein distance through quantiles with:

$$W_2^2(\mu_1, \mu_2) = \int_0^1 |q_{\mu_1}(r) - q_{\mu_2}(r)|^2 dr \tag{19}$$

Moreover, the optimal transport plan is given by:

$$\pi = (q_{\mu_1}, q_{\mu_2})_{\#}\lambda_{[0,1]} \tag{20}$$

where $\lambda_{[0,1]}$ is the Lebesgue measure on $[0,1]$. The transport plan can be stored efficiently using two vectors of size $(n + m - 1)$ (see [53] Prop 3.4).

Following [53, Remark 9.6], one can define the quantile function related to the Wasserstein mean by :

$$q_{\mu^{1\to 2}} = \frac{1}{2}q_{\mu_1} + \frac{1}{2}q_{\mu_2}. \tag{21}$$

Now, let $\mu_1 \in \mathcal{P}_2^n(\mathbb{R}^d)$ and $\mu_2 \in \mathcal{P}_2^m(\mathbb{R}^d)$. Let $\mu_\theta^{1\to 2}$ be the Wasserstein mean of the projected distributions on $\theta$. Finally let $\pi^{\theta\to 1}$ denote the transport plan from $\mu_\theta^{1\to 2}$ to $\mu_1$ and $\pi^{\theta\to 2}$ be the transport plan from $\mu_\theta^{1\to 2}$ to $\mu_2$. Following the construction of [4, Sec. 9.2], we shall introduce a multi marginal plan defined as:

$$\pi \in \mathcal{P}_2(\mathbb{R}^d \times \mathbb{R}^d \times \mathbb{R}^d) \text{ s.t. } P_{\#}^{12}\pi = \pi^{\theta\to 1} \text{ , } P_{\#}^{13}\pi = \pi^{\theta\to 2} \text{ and } \pi \in \Pi(\mu_\theta^{1\to 2}, \mu_1, \mu_2) \tag{22}$$

where $P^{12} : (\mathbb{R}^d)^3 \to (\mathbb{R}^d)^2$ projects to the first two coordinates and $P^{13}$ projects to the coordinates 1 and 3. In particular, $P_{\#}^{12}\pi$ is the projection of $\pi$ on its 2 first marginals and $P_{\#}^{13}\pi$ on the first and 3rd marginal. Similarly to the 2-marginal transport plan we defined $\Pi(\mu_\theta^{1\to 2}, \mu_1, \mu_2) = \{\pi \in \mathcal{P}_2(\mathbb{R}^d \times \mathbb{R}^d \times \mathbb{R}^d) \text{ s.t. } \pi(A \times \mathbb{R}^d \times \mathbb{R}^d) = \mu_\theta^{1\to 2}(A) \text{ , } \pi(\mathbb{R}^d \times A, \times\mathbb{R}^d) = \mu_1(A) \text{ and } \pi(\mathbb{R}^d \times \mathbb{R}^d \times A) = \mu_2(A),$ $\forall A$ measurable set of $\mathbb{R}^d\}$:

The generalized barycenter $\mu_{g,\theta}^{1\to 2}$ is then defined as:

$$\mu_{g,\theta}^{1\to 2} = \left(\frac{1}{2}P^2 + \frac{1}{2}P^3\right)_{\#} \pi \tag{23}$$

where $P^i$ is the projection on the $i$-th coordinate.

We finally have all the building blocks to compute SWGG in the general case. Let remark that the complexity goes from $\mathcal{O}(dn + n\log n)$ in the $\mathcal{P}_2^n(\mathbb{R}^d)$ case to $\mathcal{O}(d(n+m) + (n+m)\log(n+m))$ in the general case.

## 7.3 Proof of Proposition 3.2

We aim to prove that $\mathrm{SWGG}_2^2(\mu_1, \mu_2, \theta)$ is an upper bound of $W_2^2(\mu_1, \mu_2)$ and that $\mathrm{SWGG}(\mu_1, \mu_2, \theta)$ is a distance $\forall \theta \in \mathbb{S}^{d-1}, \mu_i \in \mathcal{P}_2^n(\mathbb{R}^d), i = 1, 2$.

**Distance.** Note that this proof will be derived for the alternative definition of SWGG in supp. 10.8. Let $\mu_1 = \frac{1}{n}\sum \delta_{\boldsymbol{x}_i}, \mu_2 = \frac{1}{n}\sum \delta_{\boldsymbol{y}_i}, \mu_3 = \frac{1}{n}\sum \delta_{\boldsymbol{z}_i}$ be in $\mathcal{P}_2(\mathbb{R}^d)$, let $\theta \in \mathbb{S}^{d-1}$. We note $\sigma$ (resp. $\tau$ and $\pi$) the permutation such that $\langle \boldsymbol{x}_{\sigma(1)}, \theta\rangle \leq ... \leq ...\langle \boldsymbol{x}_{\sigma(n)}, \theta\rangle$ (resp. $\langle \boldsymbol{y}_{\tau(1)}, \theta\rangle \leq ... \leq ...\langle \boldsymbol{y}_{\tau(n)}, \theta\rangle$ and $\langle \boldsymbol{z}_{\pi(1)}, \theta\rangle \leq ... \leq ...\langle \boldsymbol{z}_{\pi(n)}, \theta\rangle$).

*Non-negativity and finite value.* From the $\ell_2$ norm, it is derived

*Symmetry.* $\mathrm{SWGG}_2^2(\mu_1, \mu_2, \theta) = \frac{1}{n}\sum_i \|\boldsymbol{x}_{\sigma(i)} - \boldsymbol{y}_{\tau(i)}\|_2^2 = \frac{1}{n}\sum_i \|\boldsymbol{y}_{\tau(i)} - \boldsymbol{x}_{\sigma(i)}\|_2^2 = \mathrm{SWGG}_2^2(\mu_2, \mu_1, \theta)$

*Identity property.* From one side, $\mu_1 = \mu_2$ implies that $\langle \boldsymbol{x}_i, \theta\rangle = \langle \boldsymbol{y}_i, \theta\rangle, \forall 1 \leq i \leq n$ and that $\sigma = \tau$, which implies $\mathrm{SWGG}_2^2(\mu_1, \mu_2, \theta) = 0$.

From the other side, $\mathrm{SWGG}_2^2(\mu_1, \mu_2, \theta) = 0 \implies \frac{1}{n}\sum \|\boldsymbol{x}_{\sigma(i)} - \boldsymbol{y}_{\tau(i)}\|_2^2 = 0 \implies \boldsymbol{x}_{\sigma(i)} = \boldsymbol{y}_{\tau(i)},$ $\forall 1 \leq i \leq n \implies \mu_1 = \mu_2$.

*Triangle Inequality.* We have $\text{SWGG}_2(\mu_1,\mu_2,\theta) = \left(\frac{1}{n}\sum_i \|\boldsymbol{x}_{\sigma(i)} - \boldsymbol{y}_{\tau(i)}\|_2^2\right)^{1/2}$

$\leq \left(\sum_i \|\boldsymbol{x}_{\sigma(i)} - \boldsymbol{z}_{\pi(i)}\|_2^2 + \sum_i \|\boldsymbol{z}_{\pi(i)} + \boldsymbol{y}_{\tau(i)}\|_2^2\right)^{1/2} \leq \left(\sum_i \|\boldsymbol{x}_{\sigma(i)} - \boldsymbol{z}_{\pi(i)}\|_2^2\right)^{1/2} +$

$\left(\sum_i \|\boldsymbol{z}_{\pi(i)} + \boldsymbol{y}_{\tau(i)}\|_2^2\right)^{1/2} = \text{SWGG}_2(\mu_1,\mu_3,\theta) + \text{SWGG}_2(\mu_3,\mu_2,\theta)$

**Upper Bound** The fact that min-$\text{SWGG}_2^2$ in an upper bound of $W_2^2$ comes from the sub-optimality of the permutations $\sigma_\theta, \tau_\theta$. Indeed, they induce a one-to-one correspondence $\boldsymbol{x}_{\sigma_\theta(i)} \to \boldsymbol{y}_{\tau_\theta(i)}$ $\forall 1 \leq i \leq n$. This correspondence corresponds to a transport map $T^\theta$ such that $T^\theta_{\#}\mu_1 = \mu_2$. Since $W_2^2 = \inf_{T \text{ s.t. } T_{\#}\mu_1 = \mu_2} \frac{1}{n}\sum \|\boldsymbol{x} - T(\boldsymbol{x})\|_2^2$ we necessarily have $W_2^2 \leq$ min-$\text{SWGG}_2^2$.

**Equality** The equality $W_2^2 = $ min-$\text{SWGG}_2^2$ whenever $d > 2n$ comes from the fact that all the permutations are within the range of SWGG. In particular minimizing SWGG is equivalent to solve the Monge problem. We refer to Supp. 11.1 for more details.

### 7.4 Difference between max-SW and min-SWGG

Herein, we give an example where the selected vectors $\theta$ for max-SW and min-SWGG differ.

Let $\mu_1, \mu_2 \in \mathcal{P}(\mathbb{R}^2)$ be an empirical sampling of $\mathcal{N}(m_1, \Sigma_1)$ and of $\mathcal{N}(m_2, \Sigma_2)$ with $m_1 = \begin{pmatrix} -10 \\ 0 \end{pmatrix}$, $m_2 = \begin{pmatrix} 10 \\ 0 \end{pmatrix}$, $\Sigma_1 = \begin{pmatrix} 1 & 0 \\ 0 & 11 \end{pmatrix}$ and $\Sigma_2 = \begin{pmatrix} 2 & 0 \\ 0 & 2 \end{pmatrix}$.

Since these two distributions are far away on the $x$-coordinate, max-SW will catch this difference between the means by selecting $\theta \approx \begin{pmatrix} 1 \\ 0 \end{pmatrix}$. Indeed, the projection on the $x$-coordinate represents the largest 1D WD.

Conversely, min-SWGG selects the pivot measure to be supported on $\theta \approx \begin{pmatrix} 1 \\ 0 \end{pmatrix}$ that separates the two distributions. Indeed, this direction better captures the geometry of the 2 distributions, delivering permutations that are well grounded to minimize the transport cost.

Fig. 6 illustrates that difference between max-SW and min-SWGG.

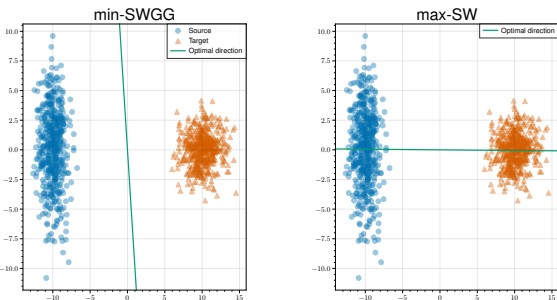

Figure 6: Optimal $\theta$ for max-SW and min-SWGG

### 7.5 From permutations to transport map

In this section we provide the way of having a transport map from permutations.

Let $\mu_1, \mu_2 \in \mathcal{P}_2^n(\mathbb{R}^d)$, let $\theta^* \in \arg\min \text{SWGG}$ and let $\sigma_{\theta^*}, \tau_{\theta^*}$ the associated permutations. The associated map must be $T(\boldsymbol{x}_{\sigma(i)}) = \boldsymbol{y}_{\tau(i)}$ $\forall 1 \leq i \leq n$. In the paper, we formulate the associated transport map as:

$$T(\boldsymbol{x}_i) = \boldsymbol{y}_{\tau_{\theta^*}^{-1}(\sigma_{\theta^*}(i))}, \quad \forall 1 \leq i \leq n. \tag{24}$$

Moreover, the matrix representation of $T$ is given by:

$$T_{ij} = \begin{cases} \frac{1}{n} & \text{if } \sigma(i) = \tau(j) \\ 0 & \text{otherwise} \end{cases} \tag{25}$$

## 7.6 Examples of Transport Plan

Fig. 7 illustrates two instances of the transport plan obtained via min-SWGG. Even though these transport plans are not optimal, they were able to capture the overall structure of the true optimal transport plans.

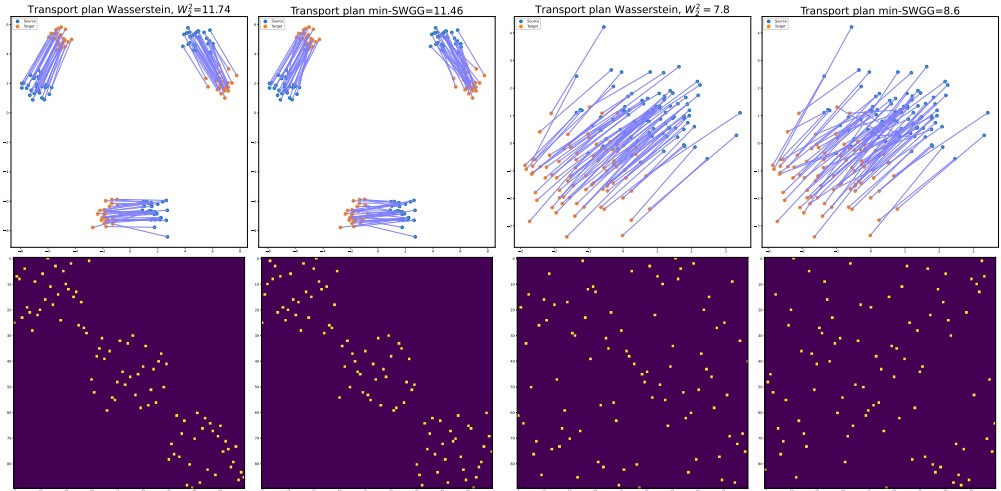

Figure 7: Example of transports plan given by Wasserstein (left and middle-right) and min-SWGG (middle left and right). Transport plan distribution (top) and transport matrix (bottom).The relative distances between source and target are given in the title.

The first example shows that the OT plan by min-SWGG exhibits a "block" structure, and thus approximates well the true Wasserstein distance. The second example shows that even in a context of superimposed distribution the "general transport direction" in min-SWGG is representative of that of the optimal transport map.

## 8 Background on Wasserstein Generalized Geodesics

We introduce some concepts related the Wasserstein generalized geodesics in Sec. 4.1. In this section, we provide more details about these geodesics in order to provide a wider view on this theory.

In the following definitions, we do not address the issue of uniqueness of the geodesics. However this is not a problem in our setup since we focus our study on pivot measure with $n$-atoms $\nu \in \mathcal{P}_2^n(\mathbb{R}^d)$. In this case, we have uniqueness of the $\nu$-based Wasserstein distance [47].

**Wasserstein generalized geodesics**  As mentioned in Sec. 4.1, Wasserstein generalized geodesics rely on a pivot measure $\nu \in \mathcal{P}_2^n(\mathbb{R}^d)$ to transport $\mu_1$ to $\mu_2$. Indeed, one can leverage the optimal transport maps $T^{\nu \to \mu_1}$ and $T^{\nu \to \mu_2}$ to construct a curve linking $\mu_1$ to $\mu_2$. The generalized geodesic with pivot measure $\nu$ is defined as:

$$\mu_g^{1 \to 2}(t) \stackrel{\text{def}}{=} ((1-t)T^{\nu \to \mu_1} + tT^{\nu \to \mu_2})_{\#}\nu \qquad \forall t \in [0,1]. \tag{26}$$

The generalized Wasserstein mean refers to the middle of the geodesic, i.e. when $t = 0.5$ and has been denoted $\mu_g^{1 \to 2}$.

Intuitively, the optimal transport maps between $\nu$ and $\mu_i, i = 1, 2$ give rise to a sub-optimal transport map between $\mu_1$ and $\mu_2$ through:

$$T_\nu^{1 \to 2} \stackrel{\text{def}}{=} T^{\nu \to \mu_2} \circ T^{\mu_1 \to \nu} \quad \text{with} \quad (T_\nu^{1 \to 2})_{\#}\mu_1 = \mu_2. \tag{27}$$

$T_\nu^{1\to2}$ links $\mu_1$ to $\mu_2$ via the generalized geodesic:
$$\mu_g^{1\to2}(t) = ((1-t)Id + tT_\nu^{1\to2})_{\#}\mu_1. \tag{28}$$

We recall here the $\nu$-based Wasserstein distance induced by $T_\nu^{1\to2}$ and introduced in Eq. (13).

**Definition 8.1.** The $\nu$-based Wasserstein distance [21, 47] is defined as:
$$W_\nu^2(\mu_1,\mu_2) \overset{\text{def}}{=} \int_{\mathbb{R}^d} \|\boldsymbol{x} - T_\nu^{1\to2}(\boldsymbol{x})\|_2^2 d\mu_1(\boldsymbol{x}) \tag{29}$$
$$= \int_{\mathbb{R}^d} \|T^{\nu\to\mu_1}(\boldsymbol{z}) - T^{\nu\to\mu_2}(\boldsymbol{z})\|_2^2 d\nu(\boldsymbol{z}). \tag{30}$$

Moreover, this new notion of geodesics comes with an inequality, which is of the opposite side to Eq. (3):
$$W_2^2(\mu_g^{1\to2}(t),\nu) \le (1-t)W_2^2(\mu_1,\nu) + tW_2^2(\nu,\mu_2) - t(1-t)W_2^2(\mu_1,\mu_2). \tag{31}$$

The parallelogram law is not respected but straddles with eq. (3) and eq. (31). We refer to Figure 8 for an intuition behind positive curvature [51], parallelogram law and generalized geodesics.

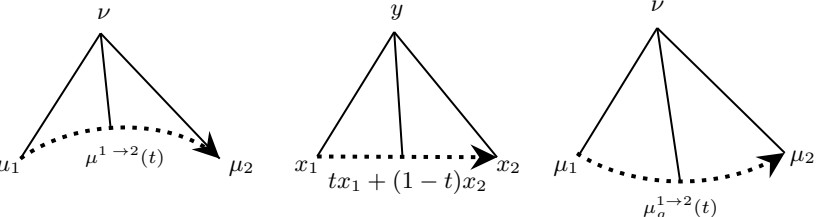

Figure 8: Geodesic $(tId + (1-t)T^{1\to2})_{\#}\mu_1$ and generalized geodesic $(tId + (1-t)T_\nu^{1\to2})_{\#}\mu_1$ in Wasserstein space (Left and Right) in dashed line and parallelogram law in $\mathbb{R}^d$ (middle).

Setting $t = 0.5$ in Eq. (31) and reordering the term gives:
$$W_2^2(\mu_1,\mu_2) \le 2W_2^2(\mu_1,\nu) + 2W_2^2(\nu,\mu_2) - 4W_2^2(\mu_g^{1\to2},\nu). \tag{32}$$

Moreover one can remark that:
$$W_\nu^2(\mu_1,\mu_2) = 2W_2^2(\mu_1,\nu) + 2W_2^2(\nu,\mu_2) - 4W_2^2(\mu_g^{1\to2},\nu) \tag{33}$$

In particular situations $W_\nu^2$ and $W_2^2$ coincide. It is the case for 1D distributions where the Wasserstein space is known to be flat [4]. In that case, the Wasserstein mean and the generalized Wasserstein mean are the same.

**Multi-marginal** Another formulation of the $\nu$-based Wasserstein distance is possible through the perspective of multi-marginal OT [4]. Let $\Pi(\mu_1,\mu_2,\nu) = \{\pi \text{ s.t. } P_\#^{12}\pi = \pi^{1\to2} , P_\#^{13}\pi = \pi^{1\to\nu} \text{ and } P_\#^{23}\pi = \pi^{2\to\nu}\}$, where $P^{ij}$ is the projection onto the coordinates $i,j$. Let also $\Pi^*(\mu_i,\nu)$ be the space of optimal transport maps between $\mu_i$ and $\nu$. We have:
$$W_\nu^2(\mu_1,\mu_2) = \inf_{\pi\in\Pi(\mu_1,\mu_2,\nu) \text{ s.t. } P_\#^{i3}\pi\in\Pi^*(\mu_i,\nu) \ i=1,2} \int_{\mathbb{R}^d} \|\boldsymbol{x}-\boldsymbol{y}\|_2^2 d\pi(\boldsymbol{x},\boldsymbol{y}) \tag{34}$$

Equation (34) expresses the fact that we select the optimal plan from $\Pi(\mu_1,\mu_2,\nu)$ which is already optimal for $\Pi(\mu_i,\nu)$. Mathematically, this minimization is not a multi-marginal problem, since the optimal plan is supposed to be already optimal for some coordinate.

The set $\{\pi \in \Pi(\mu_1,\mu_2,\nu) \text{ s.t. } P_\#^{i3}\pi \in \Pi^*(\mu_i,\nu) \ i=1,2\}$ is never empty, i.e. there is always existence of $\pi_\nu^{1\to2}$ (thanks to the gluing lemma [63], page 23). Moreover, in situations where it is a singleton, there is uniqueness of $\pi_\nu^{1\to2}$. Uniqueness is an ingredient which overpasses the selection of a final coupling and comes with additional result.

**Lemma 8.2** (Lemma 6 [47]). Whenever $\{\pi \in \Pi(\mu_1,\mu_2,\nu) \text{ s.t. } P_\#^{i3}\pi \in \Pi^*(\mu_i,\nu) \ i=1,2\}$ is a singleton, $W_\nu^2$ is a proper distance. It is a semi-distance otherwise.

Notably, 1D pivot measure was studied in [35] to ensure a dendritic structure of the distributions along the geodesic.

# 9 Related Works

In this section we highlight the fact that several upper approximations of $W_2^2$ are in the framework of generalized geodesics. The differences lay in the choice of the pivot measure $\nu$.

**Factored Coupling.** In [29], the authors impose a low rank structure on the transport plan by factorizing the couplings through a pivot measure $\nu$ expressed as the $k$-Wasserstein mean between $\mu_1$ and $\mu_2$ ($k \leq n$). It is of particular interest since whenever the pivot distribution is the Wasserstein mean between $\mu_1$ and $\mu_2$, $W_\nu^2$ and $W_2^2$ coincide.

Factored coupling results in a problem of computing the $k$-Wasserstein mean ($\mu^{1 \to 2}$) followed by solving two OT problems between the clustered Wasserstein mean and the two input distributions ($W_2^2(\mu_1, \mu^{1 \to 2})$ and $W_2^2(\mu^{1 \to 2}, \mu_2)$). Even though the OT problems are smaller, they are still expensive in practice.

Moreover, in this scenario, the uniqueness of the OT plan $T_\nu^{1 \to 2}$ is not ensured. It appears that [29] chooses the most entropic transport plan, i.e. simply $T_\nu^{1 \to 2} = T^{\mu^{1 \to 2} \to \mu_2} \circ T^{\mu_1 \to \mu^{1 \to 2}}$.

**Subspace Detours.** From a statistical point of view, it is beneficial to consider optimal transport on a lower dimensional manifold [66]. In [44], authors compute an optimal transport plan $T^{\mu_1^E \to \mu_2^E}$ between projections on a lower linear subspace $E$ of $\mu_1$ and $\mu_2$, i.e. $\mu_i^E = P_E \# \mu_i$, where $P_E$ is the linear projection on $E$. They aimed at leveraging $T^{\mu_1^E \to \mu_2^E}$ to construct a sub-optimal map $T_E^{1 \to 2}$ between $\mu_1$ and $\mu_2$.

The problem can be recast as a generalized geodesic problem with $\nu$ being the Wasserstein mean of $\mu_1^E$ and $\mu_2^E$ embedded in $\mathbb{R}^d$. Once again, uniqueness of $T_\nu^{\mu_1 \to \mu_2}$ is not guaranteed, authors provide two ways of selecting the map, namely Monge-Knothe and Monge-Independent lifting.

Subspace detours result in a problem where one needs to select a linear subspace $E$ (which is a non convex procedure), compute an optimal transport between $\mu_1$ and $\mu_2$ (in $\mathcal{O}(n^3 \log n)$ whenever $\dim(E) > 1$) and reconstruct $T_E^{\mu_1 \to \mu_2}$.

**Linear Optimal Transport (LOT).** Given a set of distributions $(\mu_i)_{i=1}^m \in \mathcal{P}_2(\mathbb{R}^d)^m$, LOT [65] embeds the set of distributions into the $L^2(\nu)$-space by computing the OT of each distribution to the pivot distribution. Mathematically, it computes $T^{\nu \to \mu_i} \forall 1 \leq i \leq m$ and lies on estimating $W_2^2(\mu_i, \mu_j)$ with $W_\nu^2(\mu_i, \mu_j)$ through eq. (13).

In LOT, the pivot measure $\nu$ was chosen to be the average of the input measures [65], the Lebesgue measure on $\mathbb{R}^d$ [41] or an isotropic Gaussian distribution [43].

Instead of computing $\binom{m}{2}$ expensive Wasserstein distances, it resorts only on $m$ Wasserstein distances between $(\mu_i)_i^m$ and $\nu$. While significantly reducing the computational cost when several distributions are at stake, it does not allow speeding up the computation when only two distributions are involved.

## 9.1 Linear Optimal Transport with shift and scaling

In this section, we recall the result from [43]. The theorem states that the $\nu$-based approximation is very close to WD whenever $\mu_1$, $\mu_2$ are continuous distributions which are very close to be shift and scaling of each other. It can applies to a continuous version of SWGG, however it works with discrete measures in the particular case of equality between $W_\nu^2$ and $W_2^2$.

**Theorem 9.1** (Theorem 4.1 [43]). Let $\Lambda = \{S_a \text{ (shift) }, a \in \mathbb{R}^d\} \cup \{R_c \text{ (scaling) }, c \in \mathbb{R}\}$, $\Lambda_{\mu,R} = \{h \in \Lambda \text{ s.t. } \|h\|_\mu \geq R\}$ and $G_{\mu,R,\epsilon} = \{g \in L^2(\mathbb{R}^d, \mu) \text{ s.t. } \exists h \in \Lambda_{\mu,R} \text{ s.t. } \|g - h\|_\mu \leq \epsilon\}$

Let $\nu, \mu \in \mathcal{P}_2(\mathbb{R}^d)$, with $\mu, \nu \ll \lambda$ (the Lebesgue measure). Let $R > 0, \epsilon > 0$

- For $g_1, g_2 \in G_{\mu,R,\epsilon}$ and $\nu = \lambda$ on a convex compact subset of $\mathbb{R}^d$, we have:

$$W_\nu(g_{1\#}\mu, g_{2\#}\mu) - W_2(g_{1\#}\mu, g_{2\#}\mu) \leq C\epsilon^{\frac{2}{15}} + 2\epsilon \tag{35}$$

- If $\mu$ and $\nu$ satisfy the assumption of Caffarelli's regularity theorem [14], then for $g_1, g_2 \in G_{\mu,R,\epsilon}$, we have:

$$W_\nu(g_{1\#}\mu, g_{2\#}\mu) - W_2(g_{1\#}\mu, g_{2\#}\mu) \leq \overline{C}\epsilon^{1/2} + C\epsilon \tag{36}$$

where $C, \overline{C}$ depdends on $\nu, \mu$ and $R$.

# 10 Proofs and other results related to Section 4

## 10.1 Proof of Proposition 4.2: equivalence between the two formulations of SWGG

In this section, we prove that the two definitions of SWGG in Def. 3.1 and Prop. 4.2 are equivalent. Let $\theta \in \mathbb{S}^{d-1}$ be fixed.

From one side in Def. 3.1, we have:

$$\mathrm{SWGG}_2^2(\mu_1, \mu_2, \theta) \overset{\mathrm{def}}{=} \frac{1}{n}\sum_i \|\boldsymbol{x}_{\sigma_\theta(i)} - \boldsymbol{y}_{\tau_\theta(i)}\|_2^2 \tag{37}$$

where $\sigma_\theta$ and $\tau_\theta$ are the permutations obtained by sorting $P_\#^\theta \mu_1$ and $P_\#^\theta \mu_2$.

From the other side we note $D(\mu_1, \mu_2, \theta)$ the quantity:

$$D(\mu_1, \mu_2, \theta) \overset{\mathrm{def}}{=} 2W_2^2(\mu_1, \mu_\theta^{1\to2}) + 2W_2^2(\mu_\theta^{1\to2}, \mu_2) - 4W_2^2(\mu_{g,\theta}^{1\to2}, \mu_\theta^{1\to2}). \tag{38}$$

We want to prove that $\mathrm{SWGG}_2^2(\mu_1, \mu_2, \theta) = D(\mu_1, \mu_2, \theta), \quad \forall \mu_1, \mu_2 \in \mathcal{P}_2^n(\mathbb{R}^d)$ and $\theta \in \mathbb{S}^{d-1}$.

Eq. (13) in the main paper states that $D(\mu_1, \mu_2, \theta)$ is equivalent to $\int_{\mathbb{R}^d} \|\boldsymbol{x} - T_{\mu_\theta^{1\to2}}^{1\to2}(\boldsymbol{x})\|_2^2 d\mu_1(\boldsymbol{x})$.

Finally, Lemma 4.6 states that the transport map $T_{\mu_\theta^{1\to2}}^{1\to2}$ is fully determined by the permutations on the line: the projections part is a one-to-one correspondence between $\boldsymbol{x}$ and $\theta\langle\boldsymbol{x}, \theta\rangle$ (resp. between $\boldsymbol{y}$ and $\theta\langle\boldsymbol{y}, \theta\rangle$). More formally $T_{\mu_\theta^{1\to2}}^{1\to2}(\boldsymbol{x}_{\sigma_\theta(i)}) = \boldsymbol{y}_{\tau_\theta(i)} \quad \forall 1 \leq i \leq n$. And thus we recover:

$$\int_{\mathbb{R}^d} \|\boldsymbol{x} - T_{\mu_\theta^{1\to2}}^{1\to2}(\boldsymbol{x})\|_2^2 d\mu_1(\boldsymbol{x}) = \frac{1}{n}\sum_i \|\boldsymbol{x}_{\sigma_\theta(i)} - \boldsymbol{y}_{\tau_\theta(i)}\|_2^2 \tag{39}$$

which concludes the proof.

## 10.2 Proof of Weak Convergence (Proposition 4.3)

We want to prove that, for a sequence of measures $(\mu_k)_{k\in\mathbb{N}} \in \mathcal{P}_2^n(\mathbb{R}^d)$, we have:

$$\mu_k \overset{\mathcal{L},2}{\longrightarrow} \mu \in \mathcal{P}_2^n(\mathbb{R}^d)) \iff \text{min-SWGG}_2^2(\mu_k, \mu) \underset{k}{\longrightarrow} 0 \tag{40}$$

The notation $\mu_k \overset{\mathcal{L},2}{\longrightarrow} \mu$ stands for the weak convergence in $\mathcal{P}_2^n(\mathbb{R}^d)$ i.e. $\int_{\mathbb{R}^d} f(\boldsymbol{x})d\mu_{(k)}(\boldsymbol{x}) \to \int_{\mathbb{R}^d} f(\boldsymbol{x})d\mu(\boldsymbol{x})$ for all continuous bounded functions $f$ and for the Euclidean distance $f(\boldsymbol{x}) = \|\boldsymbol{x}_0 - \boldsymbol{x}\|_2^2$ for all $x_0 \in \mathbb{R}^d$.

From one side, if min-$\mathrm{SWGG}_2^2(\mu_k, \mu) \to 0 \implies W_2^2(\mu_k, \mu) \to 0 \implies \mu_k \overset{\mathcal{L},2}{\longrightarrow} \mu$. The first implication is due to the fact that min-$\mathrm{SWGG}_2^2$ is an upper-bounds of $W_2^2$, the Wasserstein distance, and that WD metrizes the weak convergence.

From another side, assume $\mu_k \overset{\mathcal{L},2}{\longrightarrow} \mu$; we have for any $\theta$:

1. Let $\mu_\theta^{\mu_k\to\mu} \in \mathcal{P}_2^n(\mathbb{R}^d)$ stands for the Wasserstein mean of the projections $Q_\#^\theta \mu_k$ and $Q_\#^\theta \mu$ and let $\mu_\theta^{\mu\to\mu} = Q_\#^\theta \mu$. We have $\mu_\theta^{\mu_k\to\mu}$ converges towards (in law) to $\mu_\theta^{\mu\to\mu}$, which implies that:

$$W_2^2(\mu_k, \mu_\theta^{\mu_k\to\mu}) \underset{k}{\longrightarrow} W_2^2(\mu, \mu_\theta^{\mu\to\mu}). \tag{41}$$

2. Since $\mu \in \mathcal{P}_2^n(\mathbb{R}^d)$, we have $T^{\mu_\theta^{\mu_k \to \mu} \to \mu_k} \xrightarrow[k]{} T^{\mu_\theta^{\mu_k \to \mu} \to \mu}$ (see [22], theorem 3.2). It implies that $\mu_{g,\theta}^{\mu_k \to \mu} \xrightarrow{\mathcal{L}} \mu$ and particularly:

$$W_2^2(\mu_{g,\theta}^{\mu_k \to \mu}, \mu_\theta^{\mu_k \to \mu}) \xrightarrow[k]{} W_2^2(\mu, \mu_\theta^{\mu \to \mu}) \tag{42}$$

By combining the previous elements, we get:

$$2W_2^2(\mu_k, \mu_\theta^{\mu_k \to \mu}) + 2W_2^2(\mu_\theta^{\mu_k \to \mu}, \mu_k) - 4W_2^2(\mu_{g,\theta}^{\mu_k \to \mu}, \mu_\theta^{\mu_k \to \mu}) \xrightarrow[k]{} 2W_2^2(\mu, \mu_\theta^{\mu \to \mu})$$
$$+ 2W_2^2(\mu_\theta^{\mu \to \mu}, \mu)$$
$$- 4W_2^2(\mu, \mu_\theta^{\mu \to \mu}) = 0 \tag{43}$$

The previous relation shows that $\mu_k \xrightarrow{\mathcal{L},2} \mu$ implies $\mathrm{SWGG}_2^2(\mu_k, \mu, \theta) \xrightarrow[k]{} 0$ for any $\theta$. Hence, we can conclude that:

$$\mu_k \xrightarrow{\mathcal{L},2} \mu \implies \mathrm{min\text{-}SWGG}_2^2(\mu_k, \mu) \to 0 \tag{44}$$

This concludes the proof.

Note that when $\mu_1$ and $\mu_2$ are continuous, [41] proved that when the distributions are smooth enough (i.e. respecting the Cafarelli theorem [14]), there is a bi-Holder equivalence between the $\nu$-based Wasserstein distance and $W_2^2$. Hence, it still holds for SWGG for any $\theta \in S^{d-1}$:

$$W_2^2(\mu_1, \mu_2) \leq \mathrm{SWGG}_2^2(\mu_1, \mu_2, \theta) \leq B \times W_2^2(\mu_1, \mu_2)^{2/15} \qquad \forall \mu_i \in \mathcal{P}_2(\mathbb{R}^d) \tag{45}$$

where $B$ depends on $\mu_i, i \in \{1, 2\}, \theta$ and the dimension $d$. This bound is sufficient to prove that SWGG metrizes the weak convergence in this context. We refer to [41] for more details.

## 10.3 Proof of Translation property (Proposition 4.4)

We prove that min-$\mathrm{SWGG}_2^2$ has the same behavior w.r.t. the translation as $W_2^2$. This property is well known for Wasserstein and useful in applications such as shape matching.

Let $\mu_1, \mu_2 \in \mathcal{P}_2^n(\mathbb{R}^d)$, and let $T^u$ (resp. $T^v$) be the map $\boldsymbol{x} \mapsto \boldsymbol{x} - \boldsymbol{u}$ (resp. $\boldsymbol{x} \mapsto \boldsymbol{x} - \boldsymbol{v}$), with $\boldsymbol{u}, \boldsymbol{v}$ vectors of $\mathbb{R}^d$.

To ease the notations, let define $\tilde{\mu}_1 = T_\#^u \mu_1$ and $\tilde{\mu}_2 = T_\#^v \mu_2$.

Let remind that in the case of Wasserstein distance we have [53](Remark 2.19):

$$W_2^2(\tilde{\mu}_1, \tilde{\mu}_2) \overset{\mathrm{def}}{=} W_2^2(T_\#^u \mu_1, T_\#^v \mu_2) = W_2^2(\mu_1, \mu_2) - 2\langle \boldsymbol{u} - \boldsymbol{v}, \boldsymbol{m}_1 - \boldsymbol{m}_2 \rangle + \|\boldsymbol{u} - \boldsymbol{v}\|_2^2 \tag{46}$$

with $\boldsymbol{m}_1 = \int_{\mathbb{R}^d} \boldsymbol{x} d\mu_1(\boldsymbol{x})$ and $\boldsymbol{m}_2 = \int_{\mathbb{R}^d} \boldsymbol{x} d\mu_2(\boldsymbol{x})$.

We aim to compute min-$\mathrm{SWGG}_2^2(\tilde{\mu}_1, \tilde{\mu}_2) \overset{\mathrm{def}}{=}$ min-$\mathrm{SWGG}_2^2(T_\#^u \mu_1, T_\#^v \mu_2)$. Let express first

$$\mathrm{SWGG}_2^2(\tilde{\mu}_1, \tilde{\mu}_2) = 2W_2^2(\tilde{\mu}_1, \tilde{\mu}_\theta^{1 \to 2}) + 2W_2^2(\tilde{\mu}_2, \tilde{\mu}_\theta^{1 \to 2}) - 4W_2^2(\tilde{\mu}_{g,\theta}^{1 \to 2}, \tilde{\mu}_\theta^{1 \to 2}) \tag{47}$$

where $\tilde{\mu}_\theta^{1 \to 2}$ is the Wasserstein mean of the projections along $\theta$ of the shifted measures $\tilde{\mu}_1 = T_\#^u \mu_1$ and $\tilde{\mu}_2 = T_\#^v \mu_2$ as in Proposition 2. The generalized Wasserstein mean $\tilde{\mu}_{g,\theta}^{1 \to 2}$ is defined accordingly (see also Proposition 11).

We have:

$$W_2^2(\tilde{\mu}_1, \tilde{\mu}_\theta^{1 \to 2}) = W_2^2(\mu_1, \mu_\theta^{1 \to 2}) - 2\langle \boldsymbol{u}, \boldsymbol{m}_1 - \boldsymbol{m}_3 \rangle + \|\boldsymbol{u}\|_2^2 \tag{48}$$

where $\boldsymbol{m}_3 = \int_{\mathbb{R}^d} \boldsymbol{x} d\tilde{\mu}_\theta^{1 \to 2}(\boldsymbol{x})$.

Similarly $W_2^2(\tilde{\mu}_2, \tilde{\mu}_\theta^{1 \to 2}) = W_2^2(\mu_2, \mu_\theta^{1 \to 2}) - 2\langle \boldsymbol{v}, \boldsymbol{m}_2 - \boldsymbol{m}_3 \rangle + \|\boldsymbol{v}\|_2^2$.

Let express now the third term in eq. (47). For that we require to define the generalized Wasserstein mean $\tilde{\mu}_{g,\theta}^{1\to2}$ with pivot measure $\tilde{\mu}_{\theta}^{1\to2}$. By the virtue of eq. (11) in the main paper, we have:

$$\tilde{\mu}_{g,\theta}^{1\to2} = \left(\frac{1}{2}T^{\tilde{\mu}_{\theta}^{1\to2}\to\tilde{\mu}_1} + \frac{1}{2}T^{\tilde{\mu}_{\theta}^{1\to2}\to\tilde{\mu}_2}\right)_{\#}\tilde{\mu}_{\theta}^{1\to2} \tag{49}$$

$$= \left(\frac{1}{2}T^{\mu_{\theta}^{1\to2}\to\mu_1} + \frac{1}{2}T^{\mu_{\theta}^{1\to2}\to\mu_2} - T^{\frac{u+v}{2}}\right)_{\#}\tilde{\mu}_{\theta}^{1\to2} \tag{50}$$

$$= T_{\#}^{\frac{u+v}{2}}\left(\left(\frac{1}{2}T^{\mu_{\theta}^{1\to2}\to\mu_1} + \frac{1}{2}T^{\mu_{\theta}^{1\to2}\to\mu_2}\right)_{\#}\mu_{\theta}^{1\to2}\right) \tag{51}$$

Hence, the third term in (47) is:

$$W_2^2(\tilde{\mu}_{g,\theta}^{1\to2}, \tilde{\mu}_{\theta}^{1\to2}) = W_2^2(\mu_{g,\theta}^{1\to2}, \mu_{\theta}^{1\to2}) - 2\left\langle\frac{u+v}{2}, \frac{m_1+m_2}{2} - m_3\right\rangle + \left\|\frac{u+v}{2}\right\|_2^2 \tag{52}$$

since the mean of a Wasserstein mean is the mean of $m_1, m_2$.

Putting all together, we have:

$$\text{min-SWGG}_2^2(T_{\#}^u\mu_1, T_{\#}^v\mu_2) = \text{min-SWGG}_2^2(\mu_1, \mu_2) \quad -4\langle u, m_1 - m_3\rangle - 4\langle v, m_2 - m_3\rangle \tag{53}$$

$$+ 8\left\langle\frac{u+v}{2}, \frac{m_1+m_2}{2} - m_3\right\rangle$$
$$+ 2\|u\|_2^2 + 2\|v\|_2^2 - 4\left\|\frac{u+v}{2}\right\|_2^2$$

$$= \text{min-SWGG}_2^2(\mu_1, \mu_2) \quad +4\langle u+v, m_3\rangle \tag{54}$$
$$- 4\langle u+v, m_3\rangle - 4\langle u, m_1\rangle - 4\langle v, m_2\rangle$$
$$+ 4\langle u+v, m_1+m_2\rangle + \|u-v\|_2^2$$
$$\text{(Parallelogram law)}$$

$$= \text{min-SWGG}_2^2(\mu_1, \mu_2) \quad -2\langle u, m_1\rangle - 2\langle v, m_2\rangle + 2\langle u, m_2\rangle + 2\langle v, m_1\rangle \tag{55}$$

$$+ \|u-v\|_2^2$$
$$= \text{min-SWGG}_2^2(\mu_1, \mu_2) \quad -2\langle u-v, m_1-m_2\rangle + \|u-v\|_2^2 \tag{56}$$

## 10.4 Proof of the new closed form of the Wasserstein distance (Lemma 4.6)

We recall and prove the lemma that makes explicit a new closed form for WD. Let $\mu_1, \mu_2$ be in $\mathcal{P}_2^n(\mathbb{R}^d)$ with $\mu_2$ a distribution supported on a line whose direction is $\theta \in \mathbb{S}^{d-1}$. We have:

$$W_2^2(\mu_1, \mu_2) = W_2^2(\mu_1, Q_{\#}^{\theta}\mu_1) + W_2^2(Q_{\#}^{\theta}\mu_1, \mu_2). \tag{57}$$

Moreover, the optimal map is given by $T^{1\to2} = T^{Q_{\#}^{\theta}\mu_1\to\mu_2} \circ T^{\mu_1\to Q_{\#}^{\theta}\mu_1} = T^{Q_{\#}^{\theta}\mu_1\to\mu_2} \circ Q^{\theta}$.

Let $\mu_1, \mu_2$ be in $\mathcal{P}_2^n(\mathbb{R}^d)$ with $\mu_2$ a distribution supported on a line of direction $\theta$. We have:

$$W_2^2(\mu_1, \mu_2) = W_2^2(\mu_1, Q_{\#}^{\theta}\mu_1) + W_2^2(Q_{\#}^{\theta}\mu_1, \mu_2) \tag{58}$$

Moreover, the optimal map is given by:

$$T^{1\to2} = T^{Q_{\#}^{\theta}\mu_1\to2} \circ T^{1\to Q_{\#}^{\theta}\mu_1} = T^{Q_{\#}^{\theta}\mu_1\to2} \circ Q^{\theta} \tag{59}$$

Here $Q^{\theta}$ is given in Def. 4.1 of the paper.

The proof of the Lemma was first inspired by [13](Proposition 2.3), where authors show that $W_C^2(\mu_1, \mu_2) = W_{C^1}^2(\mu_1, \mu) + W_{C^2}^2(\mu, \mu_2)$, with $C^1, C^2$ and $C$ some cost matrices with the constraints $C_{ij} = \min_s C_{is}^1 + C_{sj}^2$.

Let $\mu_1 = \frac{1}{n}\sum \delta_{\boldsymbol{x}_i}$ and $\mu_2 = \frac{1}{n}\delta_{\overline{\boldsymbol{y}}_i}$ be in $\mathcal{P}_2^n(\mathbb{R}^d)$ with $\mu_2$ a distribution supported on a line with direction $\theta$. Let $Q_\#^\theta \mu_1 = \overline{\mu}_1 = \frac{1}{n}\sum \delta_{\overline{\boldsymbol{x}}_i} \in \mathcal{P}_2^n(\mathbb{R}^d)$. We emphasize here the fact that the atoms of $\overline{\mu}_1$ and $\mu_2$ are supported on a line are denoted by the overline symbol.

From one side, we have:

$$W_2^2(\mu_1, \mu_2) = \inf_{T^1 \text{ s.t. } T_\#^1 \mu_1 = \mu_2} \int_{\mathbb{R}^d} \|\boldsymbol{x} - T^1(\boldsymbol{x})\|_2^2 d\mu_1(\boldsymbol{x}) \tag{60}$$

$$= \inf_{T^1 \text{ s.t. } T_\#^1 \mu_1 = \mu_2} \int_{\mathbb{R}^d} (\|\boldsymbol{x} - Q^\theta(\boldsymbol{x})\|_2^2 + \|Q^\theta(\boldsymbol{x}) - T^1(\boldsymbol{x})\|_2^2) d\mu_1(\boldsymbol{x}) \tag{61}$$

$$= \int_{\mathbb{R}^d} \|\boldsymbol{x} - Q^\theta(\boldsymbol{x})\|_2^2 d\mu_1(\boldsymbol{x}) + \inf_{T^1 \text{ s.t. } T_\#^1 \mu_1 = \mu_2} \int_{\mathbb{R}^d} \|Q^\theta(\boldsymbol{x}) - T^1(\boldsymbol{x})\|_2^2 d\mu_1(\boldsymbol{x}) \tag{62}$$

$$\geq \inf_{T^2 \text{ s.t. } T_\#^2 \mu_1 = \overline{\mu}_1} \int_{\mathbb{R}^d} \|\boldsymbol{x} - T^2(\boldsymbol{x})\|_2^2 d\mu_1(\boldsymbol{x}) + \inf_{T^3 \text{ s.t. } T_\#^3 \overline{\mu}_1 = \mu_2} \int_{\mathbb{R}^d} \|\overline{\boldsymbol{x}} - T^3(\overline{\boldsymbol{x}})\|_2^2 d\overline{\mu}_1(\overline{\boldsymbol{x}}) \tag{63}$$

$$\geq W_2^2(\mu_1, \overline{\mu}_1) + W_2^2(\overline{\mu}_1, \mu_2) \tag{64}$$

Equation (61) is obtained thanks to the Pythagorean theorem since $\langle \boldsymbol{x}_i, Q^\theta(\boldsymbol{x}_i), \overline{\boldsymbol{y}}_i \rangle$ is a right triangle $\forall 1 \leq i \leq n$. The equation (64) is obtained by taking the $\inf$ of the previous first term of the previous equation.

From the other side:

$$W_2^2(\mu_1, \overline{\mu}_1) + W_2^2(\overline{\mu}_1, \mu_2) = \int_{\mathbb{R}^d} \|\overline{\boldsymbol{x}} - T^3(\overline{\boldsymbol{x}})\|_2^2 d\overline{\mu}_1(\overline{\boldsymbol{x}}) + \int_{\mathbb{R}^d} \|\overline{\boldsymbol{x}} - T^4(\overline{\boldsymbol{x}})\|_2^2 d\overline{\mu}_1(\overline{\boldsymbol{x}}) \tag{65}$$

$$= \int_{\mathbb{R}^d} \|T^3(\overline{\boldsymbol{x}}) - T^4(\overline{\boldsymbol{x}})\|_2^2 d\overline{\mu}_1(\overline{\boldsymbol{x}}) \tag{66}$$

$$= W_{\overline{\mu}_1}^2(\mu_1, \mu_2) \geq W_2^2(\mu_1, \mu_2) \tag{67}$$

Where $T^3$ and $T^4$ are the optimal plan of $W_2^2(\mu_1, \overline{\mu}_1)$ and $+W_2^2(\overline{\mu}_1, \mu_2)$. Similarly, (65) is obtained via the Pythagorean theorem. This concludes the proof.

We plot an illustration of the lemma in Figure 9.

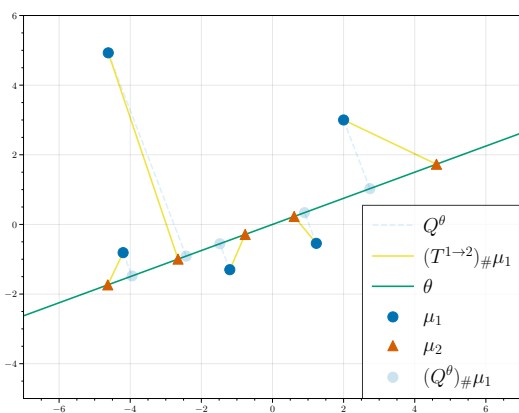

Figure 9: Closed form for Wasserstein with Pythagorus theorem

## 10.5 Details on the efficient computation of SWGG

We decompose the second formulation of SWGG. Let first remind that $Q^\theta : \mathbb{R}^d \to \mathbb{R}^d$, $\boldsymbol{x} \mapsto \theta\langle \boldsymbol{x}, \theta\rangle$ and $P^\theta : \mathbb{R}^d \to \mathbb{R}$, $\boldsymbol{x} \mapsto \langle \boldsymbol{x}, \theta\rangle$ are the projections on the subspace generated by $\theta$.

We have:

$$\text{SWGG}_2^2(\mu_1, \mu_2, \theta) = 2W_2^2(\mu_1, \mu_\theta^{1\to2}) + 2W_2^2(\mu_\theta^{1\to2}, \mu_2) - 4W_2^2(\mu_{g,\theta}^{1\to2}, \mu_\theta^{1\to2}). \tag{68}$$

First, by lemma 4.6,

$$2W_2^2(\mu_1, \mu_\theta^{1\to2}) = 2W_2^2(\mu_1, Q_\#^\theta \mu_1) + 2W_2^2(P_\#^\theta \mu_1, P_\#^\theta \mu_\theta^{1\to2}) \tag{69}$$

as $\mu_\theta^{1\to2}$'s support is on a line. Similarly,

$$2W_2^2(\mu_2, \mu_\theta^{1\to2}) = 2W_2^2(\mu_2, Q_\#^\theta \mu_2) + 2W_2^2(P_\#^\theta \mu_2, P_\#^\theta \mu_\theta^{1\to2}). \tag{70}$$

and

$$-4W_2^2(\mu_{g,\theta}^{1\to2}, \mu_\theta^{1\to2}) = -4W_2^2(\mu_{g,\theta}^{1\to2}, Q_\#^\theta \mu_{g,\theta}^{1\to2}) - 4W_2^2(P_\#^\theta \mu_{g,\theta}^{1\to2}, P_\#^\theta \mu_\theta^{1\to2}). \tag{71}$$

We notice that $2W_2^2(P_\#^\theta \mu_1, P_\#^\theta \mu_\theta^{1\to2}) + 2W_2^2(P_\#^\theta \mu_\theta^{1\to2}, P_\#^\theta \mu_2) = W_2^2(P_\#^\theta \mu_1, P_\#^\theta \mu_2)$ (as $P_\#^\theta \mu_\theta^{1\to2}$ is the Wasserstein mean between $P_\#^\theta \mu_1$ and $P_\#^\theta \mu_2$). We also notice that $-4W_2^2(P_\#^\theta \mu_{g,\theta}^{1\to2}, P_\#^\theta \mu_\theta^{1\to2}) = 0$ (it comes from the fact that the generalized Wasserstein mean is induced by the permutations on the line), we can put all together to have:

$$\mathrm{SWGG}_2^2(\mu_1, \mu_2, \theta) = 2W_2^2(\mu_1, Q_\#^\theta \mu_1) + 2W_2^2(\mu_2, Q_\#^\theta \mu_2) - 4W_2^2(\mu_{g,\theta}^{1\to2}, Q_\#^\theta \mu_{g,\theta}^{1\to2}) + W_2^2(P_\#^\theta \mu_1, P_\#^\theta \mu_2) \tag{72}$$

One can show that SWGG is divided into 3 Wasserstein distances between a distribution and its projections on a line and 1D Wasserstein problem. This results in a very fast computation of SWGG.

## 10.6  Smoothing of SWGG

In this section, we give details on the smoothing procedure of min-SWGG, an additional landscape of SWGG and its smooth counterpart $\widetilde{\mathrm{SWGG}}$ and an empirical heuristic for setting hyperparameters $s$ and $\epsilon$.

**Smoothing Procedure.**  A natural surrogate would be to add an entropic regularization within the definition of $T^{\mu_\theta^{1\to2} \to \mu_i}$, $i \in \{1,2\}$ and to solve an additional optimal transport problem. Nevertheless, it would lead to an algorithm with an $\mathcal{O}(n^2)$ complexity. Instead, we build upon the blurred Wasserstein distance [26] between two distributions $\nu_1$ and $\nu_2$:

$$B_\epsilon^2(\nu_1, \nu_2) \overset{\text{def}}{=} W_2^2(k_{\epsilon/4} * \nu_1, k_{\epsilon/4} * \nu_2)$$

where $*$ denotes the smoothing (convolution) operator and $k_{\epsilon/4}$ is the Gaussian kernel of deviation $\sqrt{\epsilon}/2$. In our case, it resorts in making $s$ copies of each sorted projections $P^\theta(\boldsymbol{x}_i)$ and $P^\theta(\boldsymbol{y}_i)$ respectively, to add a Gaussian noise of deviation $\sqrt{\epsilon}/2$ and to compute averages of sorted blurred copies $\boldsymbol{x}_{\sigma^s}^s$, $\boldsymbol{y}_{\tau^s}^s$:

$$(\widetilde{\mu_\theta^{1\to2}})_i = \frac{1}{2s} \sum_{k=(i-1)s+1}^{is} \boldsymbol{x}_{\sigma^s(k)}^s + \boldsymbol{y}_{\tau^s(k)}^s. \tag{73}$$

Further, we provide additional examples of the landscape of min-$\widetilde{\mathrm{SWGG}}(\mu_1, \mu_2)$ and discuss how to choose empirically relevant $s$ and $\epsilon$ values.

[26] has shown that the blurred WD has the same asymptotic properties as the Sinkhorn divergence, with parameter $\epsilon$ the strength of the blurring: it interpolates between WD (when $\epsilon \to 0$) and a degenerate constant value (when $\epsilon \to \infty$).

To find a minimum of Eq. (16) in the paper (i.e. $\widetilde{\mathrm{SWGG}}_2^2(\mu_1, \mu_2, \theta)$), we iterate over:

$$\theta_{t+1} = \theta_t + \eta \nabla_\theta \widetilde{\mathrm{SWGG}}_2^2(\mu_1, \mu_2, \theta)$$
$$\theta_{t+1} = \theta_{t+1}/\|\theta_{t+1}\|_2$$

where $\eta \in \mathbb{R}_+$ is the learning rate. This procedure converges towards a local minima with a complexity of $\mathcal{O}(snd + sn\log(sn))$ for each iteration. Once the optimal direction $\theta^\star$ is found, the final solution resorts to be the solution provided by $\mathrm{SWGG}_2^2(\mu_1, \mu_2, \theta^\star)$, where the induced optimal transport map is an unblurred matrix.

**Heuristic for setting the hyperparameters of** $\widetilde{\text{SWGG}}$    We here provide an heuristic for setting parameters $s$ (number of copies of each points) and $\epsilon$ (strength of the blurring). We then give an example of the behavior of $\widetilde{\text{SWGG}}$ w.r.t. these hyper parameters.

Let $\mu_1 = \frac{1}{n} \sum \delta_{\boldsymbol{x}_i}$ and $\mu_2 = \frac{1}{n} \sum \delta_{\boldsymbol{y}_i}$.

- $s \in \mathbb{N}_+$ represents the number of copies of each sample. We observe empirically that the quantity $sn$ should be large to provide a smooth landscape. It means that the $s$ values can be small when $n$ increases, allowing to keep a competitive algorithm (as the complexity depends on $ns$)

- $\epsilon \in \mathbb{R}_+$ represents the variance of the blurred copies of each sample. Empirically, $\epsilon$ should depend on the variance of the distributions projected on the line. Indeed, an $\epsilon$ very close to zero will not smooth enough the discontinuities whereas a large $\epsilon$ will give a constant landscape.

As discussed in Section 4.3, finding an optimal $\theta \in \mathbb{S}^{d-1}$ is a non convex problem and provides a discontinuous loss function. We give some examples of the landscape of $\widetilde{\text{SWGG}}$ w.r.t. different values of the hyperparameters in Fig. 10. The landscapes were computed with a set of projections $\theta$ regularly sampled with angles $\in [0, 2\pi]$.

We observe that the larger $s$, the smoother $\widetilde{\text{SWGG}}$. Additionally, raising $\epsilon$ tends to flatten $\widetilde{\text{SWGG}}$ w.r.t. $\theta$ (erasing local minima). Indeed similarly to Sinkhorn, a large $\epsilon$ blurred the transport plan and thus homogenize all the value of SWGG w.r.t. $\theta$.

Moreover, we empirically observe that the number of samples for $\mu_1$ and $\mu_2$ enforces the continuity of SWGG. We then conjecture that the discontinuities of SWGG are due to artifact of the sampling and thus the smoothing operation erases this unwanted behavior. A full investigation of this assumption is left for future work.

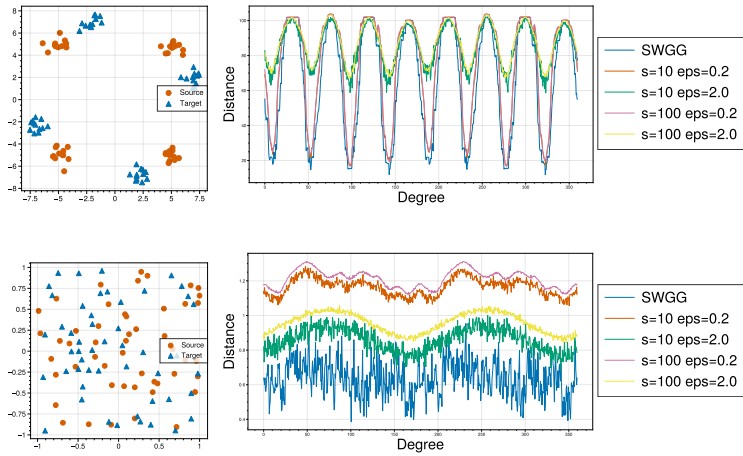

Figure 10: Non-convex landscapes for SWGG and $\widetilde{\text{SWGG}}$ with different hyper parameters.

## 10.7    Inconsequential of the pivot measure

Importantly, only the direction $\theta$ is of importance for the value of SWGG. Indeed, whenever $\nu \in \mathcal{P}_2^n(\mathbb{R}^d)$ is supported on a line of direction $\theta$, the position of the atoms is irrelevant for $W_\nu$ and the associated transport plan whenever the atoms are distinct. Despite the fact that the pivot measure is inconsequential for the value of SWGG (at $\theta$ fixed), we choose it to be $\mu_\theta^{1\to2}$. This choice is supported by the fact that $\mu_\theta^{1\to2}$ can be efficiently computed (as a 1D Wasserstein mean) and that some computation can be alleviated:

$$2W_2^2(Q_\#^\theta\mu_1, \mu_\theta^{1\to2}) + 2W_2^2(\mu_\theta^{1\to2}, Q_\#^\theta\mu_2) = W_2^2(Q_\#^\theta\mu_1, Q_\#^\theta\mu_2) \tag{74}$$

It is an important comment to derive the property of distance for SWGG; it also allows minimizing SWGG over $\theta \in \mathbb{S}^{d-1}$ without consideration for $\nu$, since any choice of $\nu$ supported on the subspace generated by $\theta$ give the same result for min-SWGG. This property of irrelevance comes from the

nature of the subspace where $\nu$ is supported, which is uni-dimensional. More formally we give the following proposition and its associated proof.

**Proposition 10.1.** Let $\mu_1, \mu_2 \in \mathcal{P}_2^n(\mathbb{R}^d)$. Let $\theta \in \mathbb{S}^{d-1}$. Let $\nu_1, \nu_2 \in \mathcal{P}_2^n(\mathbb{R}^d)$ be two pivot measures supported on a line with direction $\theta$, with disctincs atoms for each measure. We then have:

$$W_{\nu_1}^2(\mu_1, \mu_2) = W_{\nu_2}^2(\mu_1, \mu_2) \tag{75}$$

We give a proof of this proposition.

Thanks to lemma 4.6, we known that the transport map $T_\nu^{1 \to 2}$ is fully induced by the transport plan $T_\nu^{Q_\#^\theta \mu_1 \to Q_\#^\theta \mu_2}$. Let remind that $T_\nu^{Q_\#^\theta \mu_1 \to Q_\#^\theta \mu_2}$ is given by $T^{\nu \to Q_\#^\theta \mu_2} \circ T^{Q_\#^\theta \mu_1 \to \nu}$ (see equation (12)). Moreover the two optimal transport plans are obtained via the ordering permutations, i.e. let $\sigma, \tau, \pi \in \mathcal{S}(n)$ s.t:

$$\overline{\boldsymbol{x}}_{\sigma(1)} \leq ... \leq \overline{\boldsymbol{x}}_{\sigma(n)}$$
$$\overline{\boldsymbol{y}}_{\tau(1)} \leq ... \leq \overline{\boldsymbol{y}}_{\tau(n)}$$
$$\overline{\boldsymbol{z}}_{\pi(1)} \leq ... \leq \overline{\boldsymbol{z}}_{\pi(n)}$$

With $\overline{\boldsymbol{x}}_i$ being the atoms of $Q_\#^\theta \mu_1$, $\overline{\boldsymbol{y}}_i$ the atoms of $Q_\#^\theta \mu_2$ and $\overline{\boldsymbol{z}}_i$ being the atoms of $Q_\#^\theta \nu$.

One have $T^{\mu_1 \to \nu}(\boldsymbol{x}_{\sigma(i)}) = \boldsymbol{z}_{\pi(i)}$ (resp. $T^{\nu \to \mu_2}(\boldsymbol{z}_{\pi(i)}) = \boldsymbol{x}_{\tau(i)}$) $\forall 1 \leq i \leq n$. Composing these two identities gives:

$$T_\nu^{1 \to 2}(\boldsymbol{x}_{\sigma(i)}) = \boldsymbol{y}_{\tau(i)} \qquad \forall 1 \leq i \leq n \tag{76}$$

The last equation shows that $T_\nu^{1 \to 2}$ is in fact independent of $\pi$ and thus of $\nu$.

## 10.8   Proof that min-SWGG is a distance (generalized geodesic formulation)

This proof has already been established in 7.3. However we rephrase the proof in the context of generalized geodesics.

We aim to prove that $\text{SWGG}_2 = \sqrt{2W_2^2(\mu_1, \mu_\theta^{1 \to 2}) + 2W_2^2(\mu_\theta^{1 \to 2}, \mu_2) - 4W_2^2(\mu_{g,\theta}^{1 \to 2}, \mu_\theta^{1 \to 2})}$ defines a metric.

*Finite and non-negativity.* Each term of $\text{SWGG}_2^2$ is finite thus the sum of the three terms is finite. Moreover, being an upper bound of WD makes it non-negative.

*Symmetry.* We have

$$
\begin{aligned}
\text{SWGG}_2^2(\mu_1, \mu_2, \theta) &= 2W_2^2(\mu_1, \mu_\theta^{1 \to 2}) + 2W_2^2(\mu_2, \mu_\theta^{1 \to 2}) - 4W_2^2(\mu_{g,\theta}^{1 \to 2}, \mu_\theta^{1 \to 2}) \\
&= 2W_2^2(\mu_2, \mu_\theta^{1 \to 2}) + 2W_2^2(\mu_1, \mu_\theta^{1 \to 2}) - 4W_2^2(\mu_{g,\theta}^{1 \to 2}, \mu_\theta^{1 \to 2}) \\
&= \text{SWGG}_2^2(\mu_2, \mu_1, \theta).
\end{aligned}
$$

*Identity property.*
From one side, when $\mu_1 = \mu_2 \implies T^{\mu_1 \to \mu_\theta^{1 \to 2}} = T^{\mu_2 \to \mu_\theta^{1 \to 2}} = Id$, giving $\mu_{g,\theta}^{1 \to 2} = \mu_1 = \mu_2$. Thus:

$$\text{SWGG}_2^2(\mu_1, \mu_2, \theta) = 2W_2^2(\mu_1, \mu_\theta^{1 \to 2}) + 2W_2^2(\mu_1, \mu_\theta^{1 \to 2}) - 4W_2^2(\mu_1, \mu_\theta^{1 \to 2}) = 0 \tag{77}$$

From another side, $\text{SWGG}_2^2(\mu_1, \mu_2, \theta) = 0 \implies W_2^2(\mu_1, \mu_2) = 0 \implies \mu_1 = \mu_2$ (by being an upper bound of WD).

*Triangle Inequality.* We have:

$$\text{SWGG}_2^2(\mu_1, \mu_2, \theta) = \quad 2W_2^2(\mu_1, \mu_\theta^{1\to 2}) + 2W_2^2(\mu_\theta^{1\to 2}, \mu_2) - 4W_2^2(\mu_{g,\theta}^{1\to 2}, \mu_\theta^{1\to 2}) \tag{78}$$

$$= \quad 2\int_{\mathbb{R}^d} \|T_\theta^1(\boldsymbol{x}) - \boldsymbol{x}\|_2^2 d\mu_\theta^{1\to 2}(\boldsymbol{x}) + 2\int_{\mathbb{R}^d} \|T_\theta^2(\boldsymbol{x}) - \boldsymbol{x}\|_2^2 d\mu_\theta^{1\to 2}(\boldsymbol{x}) \tag{79}$$

$$- 4\int_{\mathbb{R}^d} \|T_\theta^g(\boldsymbol{x}) - \boldsymbol{x}\|_2^2 d\mu_\theta^{1\to 2}(\boldsymbol{x})$$

$$= \quad \int_{\mathbb{R}^d} \left(2\|T_\theta^1(\boldsymbol{x}) - \boldsymbol{x}\|_2^2 + 2\|T_\theta^2(\boldsymbol{x}) - \boldsymbol{x}\|_2^2 - 4\|T_\theta^g(\boldsymbol{x}) - \boldsymbol{x}\|_2^2\right) d\mu_\theta^{1\to 2}(\boldsymbol{x})$$
$$\tag{80}$$

$$= \quad \int_{\mathbb{R}^d} \|T_\theta^1(\boldsymbol{x}) - T_\theta^2(\boldsymbol{x})\|_2^2 d\mu_\theta^{1\to 2}(\boldsymbol{x}) \tag{81}$$

where, with an abuse of notation for clarity sake, $T_\theta^i$ is the optimal map between $\mu_\theta^{1\to 2}$ and $\mu_i$ and $T_\theta^g$ is the optimal map between $\mu_\theta^{1\to 2}$ and $\mu_{g,\theta}^{1\to 2}$. The last line comes from the parallelogram rule of $\mathbb{R}^d$. Thanks to Proposition 10.1 we see that SWGG is simply the $L^2(\mathbb{R}^d, \nu)$ square norm, i.e.:

$$\text{SWGG}_2^2(\mu_1, \mu_2, \theta) = \|T_\theta^1 - T_\theta^2\|_\nu^2 \overset{\text{def}}{=} \int_{\mathbb{R}^d} \|T_\theta^1 - T_\theta^2\|_2^2 d\nu \tag{82}$$

with $\nu$ being any arbitrary pivot measure of $\mathcal{P}_2^n(\mathbb{R}^d)$. And thus $\text{SWGG}_2$ is the $L^2(\mathbb{R}^d, \nu)$ norm. This observation is enough to conclude that $\text{SWGG}_2$ is a proper distance for $\theta$ fixed.

## 11 Experiment details and additional results

WD, SW, Sinkhorn, Factored coupling are computed using the `Python OT Toolbox` [28] and our code is available at `https://github.com/MaheyG/SWGG`. The Sinkhorn divergence for the point cloud matching experiment was computed thanks to the `Geomloss` package [27].

### 11.1 Behavior of min-SWGG with the dimension and the number of points

In this section, we draw two experiments to study the behavior of min-SWGG w.r.t. the dimension and to the number of points.

**Evolution with $d$**  In [20][Theorem of Section 2], authors aim at enumerate the number of permutations obtained via the projection of point clouds on a line. It appears that the number of permutations increases with the dimension. They even show that whenever $d \geq 2n$ ($2n$ being the total number of points of the problem), all the possible permutations ($n!$) are in the scope of a line. Fig. 11 depicts the number of obtainable permutations as a function of the dimension $d$, for $n$ fixed. This theorem can be applied to min-SWGG to conclude that whenever $d \geq 2n$, we have min-$\text{SWGG}_2^2 = W_2^2$.

It turns out empirically that the greater the dimension, the better the approximation of $W_2^2$ with min-SWGG (see Fig. 11) for a fixed $n$. More formally, the set of all possible transport maps is called the Birkhoff polytope and it is known that the minimum of the Monge problem is attained at the extremal points (which are exactly the set of permutations matrices, a set of $n!$ matrices in our context) [8]. The set of the transport maps in the scope of SWGG is a subset of the extremal points of the Birkhoff polytope (there are permutations matrices but not all possibilities are represented). Theoretically, the set of transport maps in the scope of SWGG is larger as $d$ grows, giving a subset that is more and more tight with the extremal points of the Birkhoff polytope. This explains that min-SWGG can benefit from higher dimension.

We plot in Fig. 11 the evolution, over 50 repetitions, of the ratio $\frac{\text{min-SWGG}(\mu_1, \mu_2)}{W_2^2(\mu_1, \mu_2)}$ with $d$, $n = 50$ and $\mu_1 \sim \mathcal{N}(1_{\mathbb{R}^d}, Id)$, $\mu_2 \sim \mathcal{N}(-1_{\mathbb{R}^d}, Id)$.

**Evolution with $n$**  Fig. 12 represents the evolution of $W_2^2(\mu_1, \mu_2)$ and min-$\text{SWGG}_2^2(\mu_1, \mu_2)$ for two distributions $\mu_1 \sim \mathcal{N}(1_{\mathbb{R}^d}, Id)$ and $\mu_2 \sim \mathcal{N}(-1_{\mathbb{R}^d}, Id)$, with $d = 4$ and a varying number of points. The results are averages over 10 repetitions.

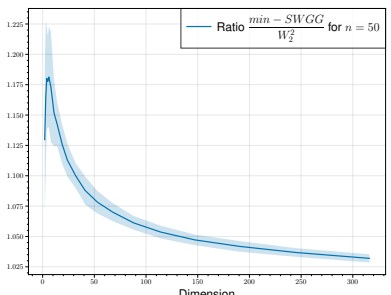
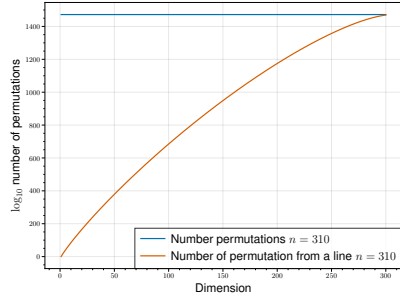

Figure 11: Evolution of $W_2^2$ and min-SWGG$_2^2$ with the dimension $d$ for isotropic Gaussian distributions (left) Number of permutations induced by a direction $\theta \in \mathbb{S}^{d-1}$ with $n = 310$ and a varying dimension (right)

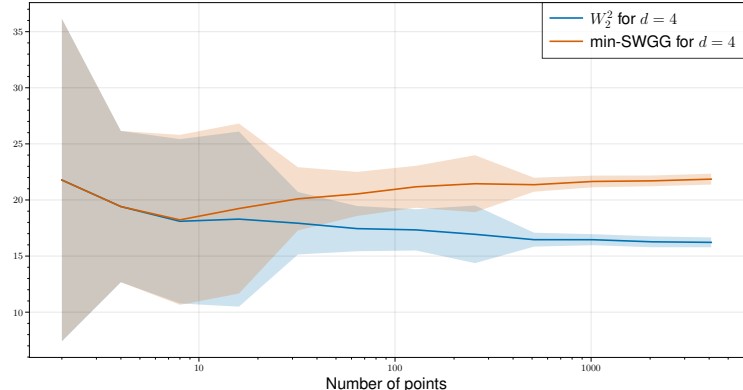

Figure 12: Evolution of $W_2^2$ and min-SWGG w.r.t. the number of points

We observe that, when $n$ is large enough, min-SWGG tends to stabilize around some constant value.

We conjecture that there may exist an upper bound for min-SWGG:

$$\text{min-SWGG}_2^2(\mu_1, \mu_2) \leq \psi(d, n, d') W_2^2(\mu_1, \mu_2) \tag{83}$$

Where $d'$ is the max of the dimensions of the distributions $\mu_1, \mu_2$ [66], and $\psi$ an unknown function.

## 11.2 Computing min-SWGG

We now provide here more details about the experimental setup of the experiments of Section 5.1.

**Choosing the optimal $\theta$** We compare three variants for choosing the optimal direction $\theta$: random search, simulated annealing and optimization (defined in Section 4.3). We choose to compare with simulated annealing since it is widely used in discrete problem (such as the travelling salesman) and known to perform well in high dimension [62] [16] [36]. We notice in Fig. 3 of the paper that the smooth version of min-SWGG is always (comparable or) better than the simulated annealing. In this experiment, we randomly sample 2 Gaussian distributions with different means and covariances matrices, whose parameters are chosen randomly. For optimizing min-SWGG, we use the Adam optimizer of Pytorch, with a fixed learning rate of $5e^{-2}$ during $100$ iterations, considering $s = 10$ and $\epsilon = 1$.

Fig. 13 provides the timings for computing the random search approximation, simulated annealing and the optimization scheme. In all cases, we recover the linear complexity of min-SWGG (blue curves) in a log space. For the computation timings we compute min-SWGG with random search

with $L = 500$, simulated annealing (green curves) with 500 iterations with a temperature scheme $(1 - \frac{k+1}{500})_{k=1}^{500}$ and the optimization scheme (considering $s = 10$ with a fixed number of iterations for the optimization scheme equals to 100).

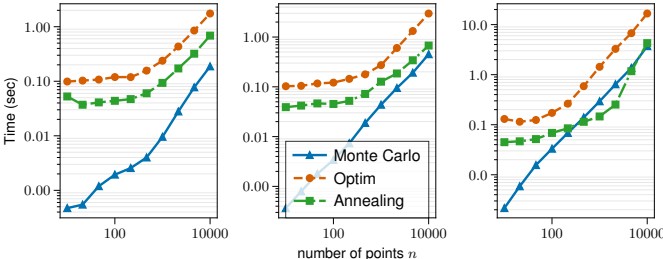

Figure 13: Considering two Gaussian distributions in dimensions $d$ equals to: 2 (left), 20 (middle), 200 (right),we compute min-SWGG with random search, simulated annealing schemes and optimization procedure and report the timings for varying number of points and fixed number of projections.

Additionally, we reproduce the same setup as in 5.1 for the SW, max-SW and PWD distance. For sake of readability we compared with min-SWGG optim and report the results in Fig. 14.

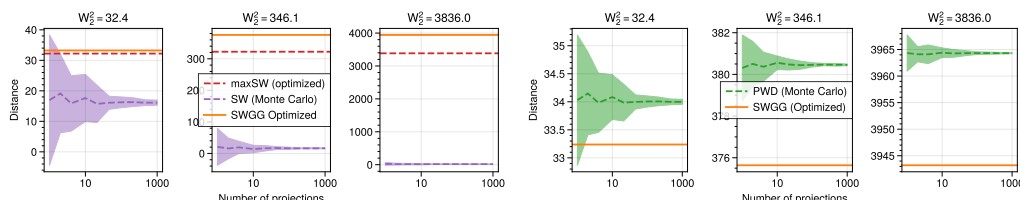

Figure 14: Comparison of min-SWGG optim with PWD (left) and with max-SW and SW (right). PWD and SW are computed with a growing number of projection

**Runtime Evaluation**   In the paper, on Fig. 3 (Right), we compare the empirical runtime evaluation on GPU for different methods. We consider Gaussian distributions in dimension $d = 3$ and we sample $n$ points per distribution with $n \in \{10^2, 10^3, 10^4, 5 \cdot 10^4, 10^5\}$. For SW Monte-Carlo and min-SWGG random search, we use $L = 200$ projections. For both max-SW and min-SWGG with optimization, we use 100 iterations with a learning rate of 1, and we fix $s = 50$ for min-SWGG. We use the official implementation of the Subspace Robust Wasserstein (SRW) with the Frank-Wolfe algorithm [52].

## 11.3   Gradient Flows

We rely on the code provided with [37] for running the experiment of Section 5.2.

We fix $n = 100$, the source distribution is taken to be Gaussian and we consider four different target measures that represent several cases: i) a 2 dimensional Gaussian, ii) a 500 dimensional Gaussian (high dimensional case), iii) 8 Gaussians (multi-modal distribution) and iv) a two-moons distribution (non-linear case).

We fix a global learning rate of $5e^{-3}$ with an Adam optimizer. For SW, PWD and SWGG (random search), we sample $L = 100$ directions. For the optimization methods max-SW, we set a learning rate of $1e^{-3}$ with a number of 100 iterations for i), iii), and iv) and 200 iterations for ii). For min-SWGG (optimization), we took a learning rate of i)$1e^{-1}$, ii)$1e^{-3}$, iii)$5e^{-2}$, and iv) $1e^{-3}$. The hyper parameters for the optimization of min-SWGG are $s = 10$ and $\epsilon = 0.5$, except for the 500-dimensional Gaussian for which we pick $\epsilon = 10$ .

Each experiment is run 10 times and shaded areas in Fig. 4 (see the main paper) represent the mean $\pm$ the standard deviation.

## 11.4 Gray scale image colorization

We now provide additional results on a pan-sharpening application to complete results provided in Section 5.3.

In pan-sharpening [64], one aims at constructing a super-resolution multi-chromatic satellite image with the help of a super-resolution mono-chromatic image (source) and low-resolution multi-chromatic image (target).

To realize this task, we choose to used a color transfer procedure, where the idea is to transfer the color palette from the target to the source image. This transfer is carry out by the optimal transport plan of the Wasserstein distance. More details on color transfer can be found in Supp. 11.6.

Additionally, we improve the relevance of the colorization by adding a proximity prior. For that, we used super pixels computed via the Watershed algorithm [48] thanks to the the `scikit-image` package [61]. Obtained high resolution colorized images of size $512 \times 512$ ($n = 262\,144$) are reported on Fig. 15.

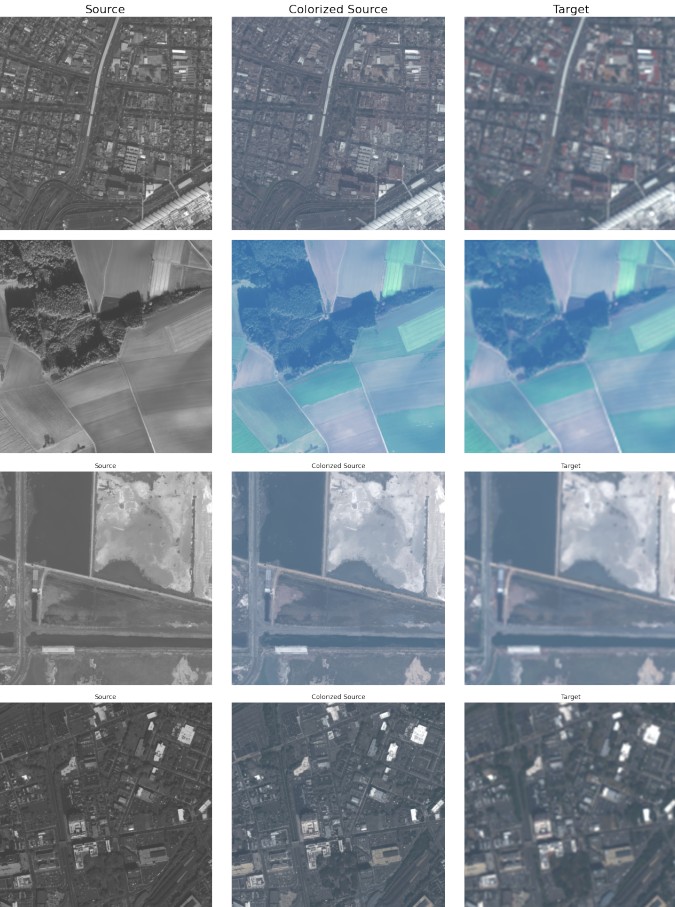

Figure 15: Source high resolution black and white image (left) Target low resolution colorful image (right) Obtained high resolution colorful image (mid).

All pan-sharpening experiments were run on the PairMax data set [64]. The hyperparameters (markers and compactness) for the watershed super-pixels are: 500, 200, 200, 200 markers (an upper bound for the number of super pixel) for each image (by order of apparition) and compactness $1e-8$ (high values result in more regularly-shaped watershed basins) for all the images.

## 11.5 Point clouds registration

We here provide additional details and results about the experiments in Section 5.4.

Authors of [9] highlighted the relevance of OT in the point clouds registration context, plugged into an Iterative Closest Point (ICP) algorithm. They leveraged the 1D partial OT without consideration for the direction of the line. Our experiment shows the importance of $\theta$: the smaller SWGG is, the better the registration.

In this experiment, having a one to one correspondence is mandatory: as such, we compare min-SWGG with a nearest neighbor assignment and the one provided by OT. Note that we do not compare min-SWGG with subspace detour [44], since: i) with empirical distributions, the reconstruction of the plan is degenerated (as it doesn't involve any computation), ii) the research of subspace can be intensive as no prior is provided.

To create the source distributions, we used random transformation $(\Omega, t) \in O(d) \times \mathbb{R}^d$ of the target domain. $\Omega$ was picked randomly from $O(d)$, the set of rotations and reflections, and $t$ has random direction with $\|t\|_2 = 5$. We also add a Gaussian noise $\mathcal{N}(0, \epsilon Id)$, with $\epsilon = 0.1$.

The ICP algorithm was run with 3 datasets with the following features: i) 500 points in 2D, ii) 3000 points in 3D, and iii) 150 000 points in 3D. min-SWGG was computed through the random search estimation with $L = 100$. A stopping criterion was the maximum number of iterations of the algorithm, which varies with the dataset $i.e.$: i) 50, ii) 100, and iii) 200 respectively. The other stopping criterion is $\|\Omega - Id\| + \|t\|_2 \leq \varepsilon$ with $\varepsilon$ chosen respectively for the datasets as follows: i) $1e^{-4}$, ii) $1e^{-2}$, and iii) $1e^{-2}$, where $(\Omega, t) \in O(d) \times \mathbb{R}^d$ is the current transformation and $\|\cdot\|$ is the Frobenius norm. All these settings were run with 50 different seeds. Results are reported in Fig. 16.

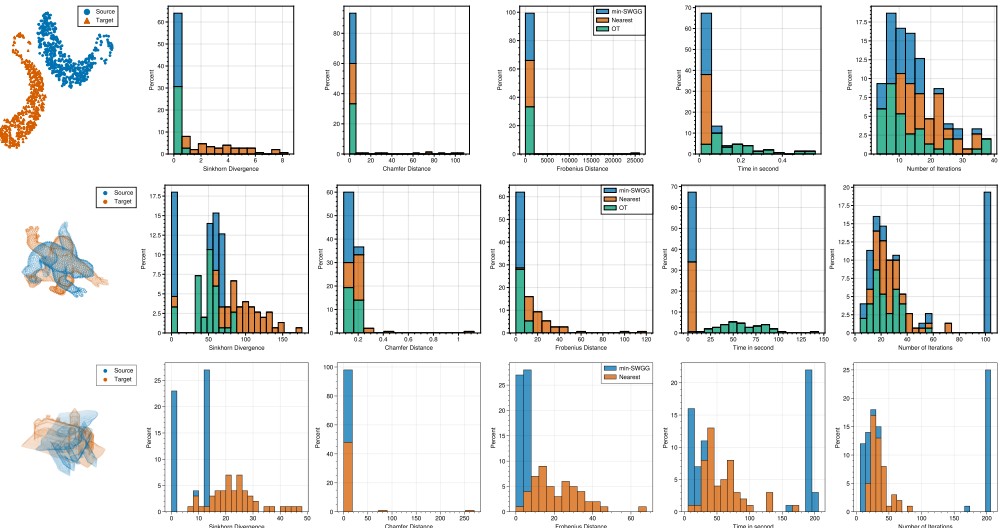

Figure 16: The three datasets (left) and the distributions of the Sinkhorn divergences, the Chamfer distance, the Frobenius distance, the timing and the number of iterations over 50 seeds (right).

From Fig. 16, one can see that for:

- $n = 500$: The registration obtained via OT is very powerful (it is fast to compute and converges toward a good solution). min-SWGG is slightly faster with better convergence result and an equivalent number of iterations. Finally the nearest neighbor does not converge to a solution closed to the target.

- $n = 3000$: registration by OT can converge poorly, moreover the timings are much higher than the competitors. min-SWGG shows efficient convergence results with an attractive computation time (order of fews seconds). We observe that the number of iterations can be very large and we conjecture that it is due to the fact that min-SWGG-based ICP can exit local minima. The nearest neighbor is fast but, most of the time, does not converge to global minima (i.e. the exact matching of the shapes).

- $n = 150000$: In this setting, OT is totally untractable (the cost matrix needs 180 GB in memory). min-SWGG shows good convergence and is most of the time very fast whenever

the number of iterations does not attain the stopping criterion. The nearest neighbor assignment is faster but only converges to local minima.

Note that, despite the fact that min-SWGG is slightly slower than the nearest neighbor, the overall algorithm can be faster due to the better quality of each iteration (min-SWGG can attain a minimum with less iterations).

In Table 3 we give additional results on the final distribution. We control the final convergence via the square Chamfer distance and the square Frobenius distance. The Square Chamfer distance is a sqaure distance between point cloud defined as:

$$d^2(X, Y) = \sum_{x \in X} \min_{y \in Y} ||x - y||_2^2 + \sum_{y \in Y} \min_{x \in X} ||x - y||_2^2 \tag{84}$$

and the Square Frobenius norm is a square distance between the transformation, defined as:

$$\text{Fr}((\Omega_{\text{real}}, t_{\text{real}}), (\Omega_{\text{estimated}}, t_{\text{estimated}})) = ||\Omega_{\text{real}} - \Omega_{\text{estimated}}||_2^2 + ||t_{\text{real}} - t_{\text{estimated}}||_2^2 \tag{85}$$

In both cases we can see that the final results for min-SWGG are much better than the results for NN and relatively closed to the results from OT.

| $n$ | 500 | 3000 | 1500 00 |
|---|---|---|---|
| NN | 11.65 | 0.20 | 6.90 |
| OT | **0.03** | 0.16 | . |
| min-SWGG | 0.08 | **0.13** | $8 \times 10^{-4}$ |

| $n$ | 500 | 3000 | 150 000 |
|---|---|---|---|
| NN | 526 | 30.04 | 21.7 |
| OT | 3.8 | 6.5 | . |
| min-SWGG | **2** | **4.5** | **6.01** |

Table 3: Square Chamfer distance (top) and Square Frobenius distance (bottom) between final transformation on the source and the target. Best values are boldfaced.

An other important aspect of ICP is that the algorithm tends to fall into local minima: the current solution is not good and further iterations do not allow a better convergence of the algorithm. We observed empirically that min-SWGG can avoid getting stuck on local minima when a reasonable number of directions $\theta$ is sampled ($L \sim 100$). We conjecture that the random search approximation is not always the ideal solution and hence may escape local minima. This may lead to a better convergence solution for min-SWGG-based ICP.

## 11.6 Color Transfer

In this section, we provide an additional experiment in a color transfer context.

We aim at adapting the color of an input image to match the color of a target one [25, 44]. This problem can be recast as an optimal transport problem where we aim at transporting the color of the source image $X$ into the target $Y$. For that, usual methods lie down on the existence of a map $T : X \rightarrow Y$. We challenge min-SWGG to this problem to highlight relevance of the obtained transport map.

Images are encoded as vector in $\mathbb{R}^{nm \times 3}$, where $n$ and $m$ are the size of the image and 3 corresponds to the number of channels (here RBG channels). We first compute a map $T_0 : X_0 \rightarrow Y_0$ between a subsample of $X$ and $Y$ of size 20000 and secondly extend this mapping to the complete distributions $T : X \rightarrow Y$ using a nearest neighbor interpolation. The subsampling step is mandatory due to the size of the images but can deteriorate the quality of the transfer.

We compare the results obtained with maps obtained from Wasserstein distance, min-SWGG with random search (100 projections), subspace detour [44] and min-SWGG (optimized). Obtained images and the associated timings are provided in fig. 17.

Figure 17 shows that min-SWGG and $W_2^2$ provide visually equivalent solutions. Since, the quality of the color transfer is dependent on the size of the subsampling: using min-SWGG permits larger

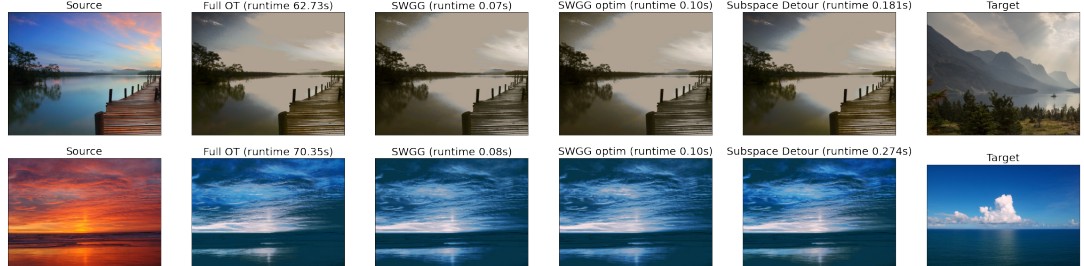

Figure 17: Color transfer of images with $W_2^2$, min-SWGG and subspace detours, with runtimes.

subsamples than $W_2^2$ and thus improves the quality of the map $T$. Moreover one can note that min-SWGG (optimized) is the fastest to compute.

We now give more details about how to perform color transfer between two distributions. The first step is to encode $n \times m$ images as $\mathbb{R}^{nm \times 3}$ vectors, with 3 channels in a RGB image. Note that $m$ and $n$ can differ for the source and target image. The second step consists of defining subsamples $X_0, Y_0$ of $X, Y$, in our case we took $\boldsymbol{X}_0, \boldsymbol{Y}_0 \in \mathbb{R}^{20000 \times d}$. We subsample the same number of points for the source and target image. In order to have a better subsampling of $\boldsymbol{X}$ and $\boldsymbol{Y}$, it is common to perform a $k$-means [34] to derive $\boldsymbol{X}_0$ and $\boldsymbol{Y}_0$ ($\boldsymbol{X}_0, \boldsymbol{Y}_0$ are then taken as centroids of the k-means algorithm). The third step is to compute $T_0 : \boldsymbol{X}_0 \to \boldsymbol{Y}_0$. We set $T$ as the optimal Monge map given by the Wasserstein distance and $T$ as the optimal map given by min-SWGG. Finally, the fourth step deals with extending $T_0 : \boldsymbol{X}_0 \to \boldsymbol{Y}_0$ to $T : \boldsymbol{X} \to \boldsymbol{Y}$. $\forall \boldsymbol{x} \in \boldsymbol{X}$. We compute the closest element $\boldsymbol{x}_0 \in \boldsymbol{X}_0$ and we pose:

$$T(\boldsymbol{x}) = T(\boldsymbol{x}_0). \tag{86}$$

More details on the overall procedure can be found in [25].

To perform the experiment, we took $L = 100$ projections for min-SWGG (random search). For min-SWGG (optimized), we fixed the following set of parameters for the gradient descent: learning rate $5e^{-2}$, number of iterations 20, number of copies $s = 10$ and $\epsilon = 1$. Regarding the subspace detour results, we used the code of [44] provided at `https://github.com/BorisMuzellec/SubspaceOT`.

Additionally, we perform color transfer without sub-sampling with the help of min-SWGG (we the same hyperparameters). This procedure is totally untractable for either $W_2^2$ and subspace detours (due to memory issues). As we mentioned before, the subsampling phase can decrease the quality of the transfer and thus min-SWGG can deliver better result than before. Result are give in Fig. 18

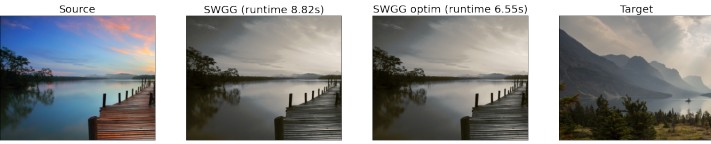

Figure 18: Color transfer of images with min-SWGG without sub-sampling.

## 11.7 Data set distance

We finally evaluate min-SWGG in an other context: computing distances between datasets. Let $\mathcal{D}_1 = \{(\boldsymbol{x}_i^1, \boldsymbol{y}_i^1)\}_{i=1}^n$ and $\mathcal{D}_2 = \{(\boldsymbol{x}_i^2, \boldsymbol{y}_i^2)\}_{i=1}^n$ be source and target data sets such that $\boldsymbol{x}_i^1, \boldsymbol{x}_i^2 \in \mathbb{R}^d$ are samples and $\boldsymbol{y}_i^1, \boldsymbol{y}_i^2$ are labels $\forall 1 \le i \le n$. In [3], the authors compare those data sets using the Wasserstein distance with the entries of the cost matrix defined as:

$$C_{ij} = \left( \|\boldsymbol{x}_i^1 - \boldsymbol{x}_j^2\|_2^2 + W_2^2(\alpha_{\boldsymbol{y}_i^1}, \alpha_{\boldsymbol{y}_j^2}) \right)^{1/2} \tag{87}$$

and the corresponding distance as:

$$OTDD(\mathcal{D}_1, \mathcal{D}_2) = \min_{P \in U} \langle C, P \rangle \tag{88}$$

where $\alpha_{\boldsymbol{y}}$ is the distribution of all samples with label $\boldsymbol{y}$, namely $\{\boldsymbol{x} \in \mathbb{R}^d | (\boldsymbol{x}, \boldsymbol{y}) \in \mathcal{D}\}$ for $\mathcal{D}$ being either $\mathcal{D}_1$ or $\mathcal{D}_2$ and $U$ is the Birkhoff polytope which encodes the marginal constraints. Notice that cost in Eq. (87) encompasses the ground distance and a label-to-label distance. This distance is appealing in transfer learning application since it is model-agnostic. However, it can be cumbersome to compute in practice since it lays down on solving multiple OT problems (to compute the cost matrix and the OTDD). To circumvent that, [3] proposed several methods to compute the cost matrix in Eq. (87). They used the Sinkhorn algorithm (in $\mathcal{O}(n^2)$) or they assumed $\alpha_{\boldsymbol{y}} \sim \mathcal{N}(m_{\boldsymbol{y}}, \Sigma_{\boldsymbol{y}})$ in order to get the WD through the Bures metric (that provides a closed form of OT for Gaussian distributions in $\mathcal{O}(d^3)$), which is still prohibitive for high dimension. We challenge min-SWGG in this context.

In this experiment, we compare the following datasets: MNIST [39], EMNIST [18], FashionMNIST [67], KMNIST [17] and USPS [32]. We rely on the code of OTDD provided at `https://github.com/microsoft/otdd`. In order to make it compliant with the min-SWGG hypothesis, we require the empirical distributions $\alpha_{\boldsymbol{y}}$ to have the same number of atoms.

Fig. 19 provides results for a batch size of $n = 40000$ samples using the Sinkhorn divergence (with a regularisation parameter of $1e^{-1}$) and for min-SWGG (optimized) on batch of size 40000. We report results for a learning rate of $1e^{-5}$, 20 iterations and $s$ and $\epsilon$ to be 1 and 0.

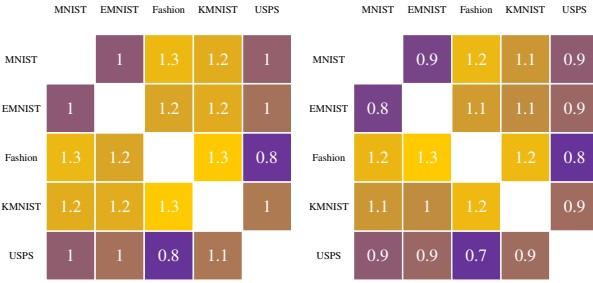

Figure 19: OTDD results ($\times 10^2$) distances for min-SWGG (left) and Sinkhorn divergence (right) for various datasets.

We check that the orders of magnitude are preserved with min-SWGG. For example OTDD(MNIST,USPS) is smaller than OTDD(MNIST,FashionMNIST) for either Sinkhorn divergence or min-SWGG as distance between labels, this validate that min-SWGG is a meaningful distance in this case scenario. Moreover in our setup, the computation cost is more expensive for Sinkhorn than for min-SWGG and totally untractable for $W_2^2$. On smaller batches (see Fig 20), the same observation can be made: min-SWGG is comparable (in term of magnitude) with $W_2^2$, Sinkhorn and the Bures approximation.

We give additional results in Fig. 20 for batches of size of $n = 2000$ samples obtained with $W_2^2$, Sinkhorn divergence (setting the entropic regularization parameter to $1e^{-1}$), the Bures approximation, min-SWGG (random search with $L = 1000$ projections) and min-SWGG (optimized, with a learning rate of $5e^{-1}$, 50 iterations, $s = 20$ and $\epsilon = 0.5$).

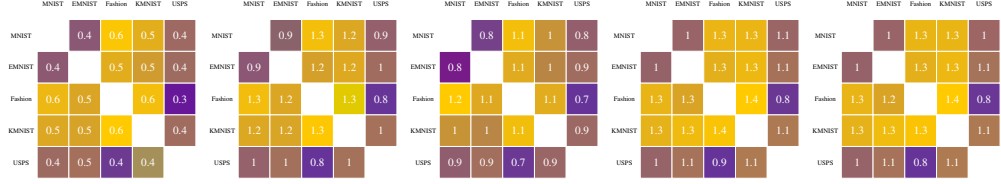

Figure 20: OTDD distance with $W_2^2$ (left), Sinkhorn (left-mid), Bures (middle) and random search min-SWGG (right-mid) and min-SWGG-optimization (right) distances between labels distribution $\times 10^2$

Note that the Figs. 19 and 20 are not symmetric (OTDD(KMNIST,FashionMNIST)$\neq$ OTDD(FashinMNIST,KMNIST)) because of the random aspect of batches.

