# OpenReview forum: "Fast Optimal Transport through Sliced Generalized Wasserstein Geodesics"
_NeurIPS.cc/2023/Conference — NeurIPS 2023 spotlight_

### Official Review · Reviewer_LrDA · 2023-06-19

**Soundness:** 3 good
**Presentation:** 2 fair
**Contribution:** 3 good
**Rating:** 8
**Confidence:** 5

**Summary:**

The paper introduces a variant of slicing of the Wasserstein distance that keeps the optimal transport map property (in particular, existence of an optimal transport map). The typical formulation of slicing loses the Monge map as each slice comes with its own map and there is no natural way to combine maps coming from different slices. As in the Wasserstein case, the variant of the sliced optimal transport distance introduced here is efficient to compute, and performs similarly (both theoretically and in practice) to the sliced Wasserstein distance.

**Strengths:**

The idea is very nice. A major advantage of sliced distances are their numerical efficiency but the big drawback is the lack of maps. By being able to define optimal transport maps the authors have removed a significant disadvantage, whilst keeping the main advantage.

**Weaknesses:**

(1) More theoretical background/justification, such as when do minimizers in Definition 4.1 exist would be helpful.
(2) The distance defined in eq. (13), since it can be written as the square root of \int_{R^d} \| T^{\nu\to\mu_1}(x) - T^{\nu\to\mu_2}(x)\|^2 d \nu(x), is the linearized Wasserstein distance. The authors should connect to the literature on linearized optimal transport distances.
(3) There were a MANY grammatical and spelling errors in the paper, including misspelling Wasserstein (line 48). The paper needs a very careful readthrough. This is my reason for scoring the paper as a 2 in presentation. I felt the explanations were otherwise quite good, and I would be happy to give a higher score once corrections have been made.

**Questions:**

(a) Connected to (1) in the weaknesses section: can you give sufficient and/or necessary conditions for existence of minimizers in Definition 4.1?
(b) What would you do if minimizers do not exist?

**Limitations:**

There was no discussion on limitations of the method, although a few directions for further work was proposed in the conclusions.

---

> ### Author Rebuttal · Authors · 2023-08-09
>
> We thank you for your review and for the positive evaluation of our paper. We now provide detailed answers to your questions.
>
> - **Existence of minimizer in Definition 4.1.** The existence of $\mu_\theta^{1\to2}$ is always guaranteed, whenever $\mu_1,\mu_2$ are two empirical probability measures. This follows from the fact that the Wasserstein mean (aka McCann interpolant with $t=1/2$) is always explicit in this case (you can refer to [1] Remark 7.1 for example). We propose to include a reference to this property in the background section of the final version.
>
> - **Link with the linearized OT.** You are correct; min-SWGG is indeed a linearized version of the Wasserstein distance where the ground measure $\nu$ is chosen carefully. We provided a brief overview of this property in Section C ('Linear Optimal Transport') of the supplementary material. We propose to strengthen this discussion with more literature on linearization. It would be interesting to elaborate on the importance of the ground measure in approximating the Wasserstein distance using the $L^2(\nu)$ distance. For instance, when the ground measure is chosen to be the Wasserstein mean of the two distributions the distance obtained through linearization equals the true Wasserstein distance.
>
> - **Grammatical and misspelling errors.** We apologize for the English errors that appear throughout the paper. We are committed to ensuring thorough proofreading in the final version. We have already addressed some issues using online tools and intend to have the paper reviewed by a native English speaker if it is accepted.
>
> [1] Peyré, G., & Cuturi, M. (2019). Computational optimal transport: With applications to data science. Foundations and Trends® in Machine Learning, 11(5-6), 355-607.)

---

> > ### Comment · Reviewer_LrDA · 2023-08-21
> >
> > I thank the authors for their response and I've increased my score as I'm happy with their response (I would particularly encourage the authors to be as thorough as possible with their grammar/spelling review, as scientifically I enjoyed reading the paper).

---

### Official Review · Reviewer_CJnA · 2023-07-02

**Soundness:** 3 good
**Presentation:** 4 excellent
**Contribution:** 3 good
**Rating:** 6
**Confidence:** 3

**Summary:**

This paper proposes an approximation to the squared Wasserstein distance between probability measures (typically required in high dimensional space) based on the sliced Wasserstein paradigm. The general idea is to project points onto a line where computing Wasserstein distances becomes significantly simpler (a typical case involves only permutations and sorting). The authors build up their main construction - min SWGG, by leveraging the properties of prior works like Max SW and projected sliced Wasserstein distance (PWD) and derive a tighter upper bound for the squared Wasserstein distance (W2) in comparison to PWD leading to the definition of min-SWGG. To put it very simply min-SWGG is closely related to the PWD, where the minimum is chosen instead of the average over all possible angles.

The authors then relate the min-SWGG formulation to Wasserstein Generalised Geodesics - which involves a third measure (called pivot) in order to approximate the actual Wasserstein distance. By choosing the pivot measure to lie on a line,  this connection enables a more tractable way to optimize for the best angle/plane to compute min-SWGG.

Experiments are demonstrated for (1.) Image Color Transfer (2.) PointCloud Matching (3.) Empirical demonstration of min SWGG optimization. The results demonstrate a proof of concept (i.e. examples showing the min-SWGG can be computed with different options quite reasonably) and some cases of good accuracy and faster computation

**Strengths:**

- Overall, I found this paper to be very comprehensive in its core formulation of min-SWGG. This paper appears to rate highly in its theoretical contributions - (1.) The definition of a new OT approximation with interesting bounds to the actual WD and PWD (2.) Connecting min SWGG to the Generalized Wasserstein Geodesic using a pivot measure that lies on a line thereby showing a path for computational tractability (3.) Proofs of metricity, weak convergence, computational and topological properties etc.
- The paper is written well with a good introduction and fairly self contained background

**Weaknesses:**

- The practical aspect of this paper is sort of underwhelming. It would have been very convincing to demonstrate on truly high dimensional applications. Most experiments are very synthetic, and despite a modest experimental section, I am still grasping for situations where min-SWGG outperforms SWD and perhaps WD in terms of computability, quality of distances and transport maps, barycenters etc.
- To this aid, despite the commentary on the use of cumulative distributions in the supplementary (enabling transport plans instead of maps), I could not place any impactful demonstration. That being said almost all of the narrative is based on matching uniform distributions having equal number of points and it's unclear how to generalize from here.
- Generally, most figures could do with a more descriptive caption. Especially Figure 4 which I am finding really hard to parse. What is the box with Gaussian 500d indicate? Which are the source and target distributions? What is the message of this experiment? (I suspect it is that with enough iterations, the optimization of min-SWGG provides lower W2 distances - please clarify)


**Questions:**

 - I am curious to visualize the difference between the transport plan obtained using min-SWGG in comparison to the optimal transport plan (i.e. corresponding to WD). This could potentially be visualized on the Pointcloud example.
- How do the other SW distances (SW, max-SW, PWD etc.) compare in figure 3 (left)?
- What is the evaluation for some other metric in the point cloud example, for eg the chamfer distance, L2 error after alignment etc. Is there a specific reason to report only the sinkhorn divergence?

**Limitations:**

See Weaknesses for technical limitations. I am not an expert, but I think this paper has no direct negative societal impact.

Overall, I am inclined positively on this paper. The new definition and approximation are derived nicely and interesting connections are made to the Wasserstein Generalised Geodesics framework. However, the practical impact (even amongst competing optimal transport techniques) is not equally impressive, and at this point unclear if it is as widely applicable. Put together, I vote a weak acceptance, with an intention to re-assess more carefully after the rebuttal.

---

> ### Author Rebuttal · Authors · 2023-08-09
>
> We thank the reviewer for the valuable comments and provide some answers to the questions and provide feeback about the reported weaknesses.
>
> - **The practical aspect of min-SWGG is underwhelming.** In the paper, we conducted several experiments in a wide set of scenarii, with the aim to stress the interest of min-SWGG. We first aimed at illustrating some properties of min-SWGG, e.g. i) study of the random search and optimization scheme (Sec. 5.1), ii) study of the runtimes wrt competitors (Sec. 5.1), iii) study of its weak convergence property (Sec. 5.2). We also provided some results in different contexts, such as color transfer and pan sharpening (Sec. 5.3, E.4 and E.6), point clouds registration (Sec. 5.4) and computing distances between datasets (Sec. E.7). We showed that the optimization scheme is relevant in high dimensional settings (5.1 and 5.2), that its computation is fast wrt competitors (Sec. 5.1 and 5.4) and that it comes with a tranport plan (which is not the case for SWD, Sec. 5.3 and 5.4). It is true that we did not elaborate on a flagship application, and we hope that future studies will allow illustrating the relevance of min-SWGG in a wider set of contexts.
>
>
> - **Visualize the transport plan of min-SWGG.** We propose to include examples of the transport plan in the ICP context in the final version. Meanwhile figure 1 (of the rebuttal's pdf) illustrates the transport plan of min-SWGG and plain OT on two examples of Gaussian or bimodal distributions.
>
> - **Evolution of SW, max-SW, PWD,.. in figure 3.** It is entirely feasible to incorporate the competitors into the figure. However the differing orders of magnitude tend to deteriorate the readabiliy of the figure. The attached Table 3 in the PDF accompanying the main author's rebuttal offers an illustration of this magnitude difference in a distinct yet related setup.
>
> - **Regarding the ICP experiment.** Tables 1 and 2 (of the rebuttal's pdf) provide additional results for the ICP experiment (with $n=500$ and $n=3000$) and show that min-SWGG performs favorably when considering square Frobenius or Chamfer distance.

---

### Official Review · Reviewer_rywq · 2023-07-05

**Soundness:** 3 good
**Presentation:** 3 good
**Contribution:** 3 good
**Rating:** 6
**Confidence:** 2

**Summary:**

This paper proposed a novel proxy of the squared Wasserstein distance called min-SWGG, which is based on Sliced-Wasserstein distance (SWD) and the projected Wasserstein distance (PWD). This quantity is also shown to be relevant to Wasserstein Generalized Geodesics with pivot measure, which makes it more interesting. The theoretical properties of min-SWGG has been discussed, including the topological properties and the relationships with Wasserstein Generalized Geodesics. The authors provide two efficient algorithms for computing min-SWGG.  The experimental results indicates better computational efficiency of the proposed algorithm compared to the state-of-the-art methods.


**Strengths:**

The paper is well written and clear, the approach is well supported by the theoretical analysis. The experiment and application seem strong and nice. The authors also made comparisons to other state-of-the-art methods that makes the paper more complete.

**Weaknesses:**

- One question about the approach in line 140: In Remark 3.3 the authors mentioned about the overall computational complexity to calculate min-SWGG by random search for L times. I wonder if there is any upperbound for it when the dimension d is high? Or is there any results showing the possible order of relation of  L and dimention d, or other quntities?

- For the second row of Figure 2, I notice that for Gaussian 500d, min-SWGG (optim) performs the best but for the left one, min-SWGG (optim) may have inferior behaviors. What might be the issue? Could you maybe elaborate on it a little bit?



**Questions:**

See weakness.



**Limitations:**

I have not found the limitations of the approach to be addressed in the paper. Maybe the authors can elaborate more on these.

---

> ### Author Rebuttal · Authors · 2023-08-09
>
> We thank the reviewer for the comment regarding the behavior of min-SWGG when $d$ varies. It is indeed an interesting question, and we do believe that min-SWGG is a particularly interesting approximation in this context. First, we recall that, when $d > 2n$, equality between $W_2^2$ and min-SWGG holds (Prop. 3.2).
>
> - **Behavior of the number of random directions with the dimension.** We expect that, as $d$ increases, a larger number of projections $L$ will be required to achieve a decent approximation. Empirically, we also observe that within a broad range of scenarios, it is sufficient to take $\mathcal{O}(L^{d-1})$ directions, similar to SWD [2] or max-SW [1]. A deeper investigation of this behavior is deferred to future work. This behavior serves as motivation for designing an optimization scheme through gradient descent.
>
> - **Variability of min-SWGG optimization with dimension in gradient flow experimentation.** You are correct: when $d$ is small, it can be observed that min-SWGG (optim) converges a little more slowly than min-SWGG (random search). This is due to the fact that min-SWGG (optim) optimizes a smooth and non-convex surrogate of min-SWGG (see eq. 16). As such, the quality of the minimum found by descent depends on the choice of the initialization. Empirically, we find that solutions provided by this surrogate may differ from those of min-SWGG (random search), even though they are close. The gradient flow experiment is run for 2000 iterations, which accounts for the final difference.
>
> [1] Deshpande, I., Hu, Y. T., Sun, R., Pyrros, A., Siddiqui, N., Koyejo, S., ... & Schwing, A. G. (2019). Max-sliced wasserstein distance and its use for gans. In Proceedings of the IEEE/CVF Conference on Computer Vision and Pattern Recognition (pp. 10648-10656).
>
> [2] Zhang, J., Ma, P., Zhong, W. and Meng, C. . Projection-based techniques for high-dimensional optimal transport problems. Wiley Interdisciplinary Reviews: Computational Statistics, 15(2):e1587, 2023.

---

> > ### Comment · Reviewer_rywq · 2023-08-16
> >
> > I would like to thank the authors for their reply. My assessment remains inclined to the positive.

---

### Official Review · Reviewer_5WTB · 2023-07-10

**Soundness:** 3 good
**Presentation:** 3 good
**Contribution:** 3 good
**Rating:** 7
**Confidence:** 4

**Summary:**

The paper proposes a new formulation and algorithm to approximate Wasserstein 2-distance (i.e., the distance of two points is the squared Euclidean one) between two distributions. Though some of the development is general, the key parts and the algorithms apply only to distributions being the sums of delta functions with the same weights. The approach builds on approaches approximating the true WD by using 1-D slices of the distributions, the distance between which can be computed cheaply as pointwise sum of squared differences of sorted distributions.

The basic Sliced WD, which approximates the true WD by the integral of the distance between such 1-D projections over all directions (eq. (5)). Its approximation is Max-Sliced WD, which replaces the integral by maximization (eq. (6)), leading to a lower bound on the true WD.

Another existing sliced formulation is Projected WD, which approximates the true WD by the integral of the pointwise sum of squared differences between the two distributions in which the points are sorted as if the distributions were projected onto the direction, over all directions (eq. (7)). The approach proposed in the manuscript replaces the integral here by minimization (eq. (9)), which is called Min-SWGG. It yields an upper bound in the true WD. This problem is non-convex, in fact discontinuous because the objective jumps with every change of either permutation (which sort the points on the projections). Therefore, it is smoothed and at the same time reformulated using the concepts from differential geometry of WD, generalized geodesics. This leads to a smooth non-convex problem, approximating the initial min-WDGG formulation. Its objective for a given direction can be evaluated cheaply: its most expensive part is the WD between a general point cloud and a point cloud on a line, which can be computed cheaply by just sorting 1-D distributions. The obtained problem is solved by gradient descent (yielding thus a local minimum).

The experiments compare the Min-SWGG, optimized either by smoothing and gradient descent or by random search or by simulated annealing, with existing (approximate) WD methods on synthetic data. Then it is illustrated on colorization of gray-scale images by OT, where computing the exact OT is intractable. Then it is demonstrated on point cloud registration by the ICP algorithm, where the point correspondence in every ICP iteration is computed either by nearest-neighbor or OT.

**Strengths:**

The problem addressed is useful in a number of applications.

Technically sound.

Clearly enough written.

**Weaknesses:**

The formulation of min-SWGG (eq. (9)) is rather straightforward. Namely, approximating the integral (7) by the minimization (9) is analogous to approximating the integral (5) by the maximization (6). In this sense, I see the novelty of the paper as rather incremental.

It is inconvenient that a lot of non-trivial information is in the supplement (I wonder if this format of presentation is desirable for NIPS). E.g., in the main paper the problem is restricted for the distributions from $P_2^n(R^d)$, i.e., unweighted averages of delta-distributions (see line 116), so that the exact WD (formula (1)) reduces to a linear assignment problem with the costs being squared Euclidean distances, for which the transport plan is a permutation. I.e., given two clouds of points in R^d with n elements, we want to find a permutation of one point cloud such that the sum of squared Euclid. distances between the points is minimum. In this restricted formulation, the min-SWGG (9) just optimizes over a subset of all possible permutations, given by sorting the projections of the distributions onto some direction. It is mentioned that this formulation can be extended to arbitrary distributions, but the details of this extension are just sketched in the supplement (section A.2). It is not clear from A.2 what would be the complexity of the algorithm in this general setting. In particular, I do not see why the transport plan (eq. (4) in A.2) should always have only $n+m-1$ nonzeros. Moreover, the experiments are done only for the restricted setting.

Details of some experiments are also often given only in the supplement. E.g., Section 5.3 has too little details to me, so that I do not understand how exactly image colorization is formulated. The additional info in the supplement is not helpful.

In experiments, Table 1 (report on point cloud registration by ICP) might be unfair to the nearest-neighbor method because the table reports Sinkhorn divergence of the registered clouds, which is close to what is optimized in min-SWGG. A better criterion to report might be the difference between the estimated and the ground-truth transformation (assuming that the latter is available, which trivially is e.g. when the target cloud is constructed by transforming the source cloud). Moreover, NN might perform well in the regime when the point clouds are initially close to each other.

Minor comments:
- The formulas for $\mu_1,\mu_2$ on line 116 are repeated on lines 122. The text could be optimized such that they appear only once.
- Denoting the distances (8) and (9) as (min-)SWGG is confusing because generalized geodesics (GG) are not needed for their definitions (they are needed only later to reformulate the definitions).

**Questions:**

Q1) In the restricted setting of unweighted discrete distributions (i.e., from $P_2^n(R^d)$), the problem (1) reduces to linear assignment problem with costs being squared Euclidean distances. This is a classical problem, for which a great number of exact and approximate algorithms have been proposed. Unfortunately, I am not that familiar with existing approximate algorithms for this problem. Did you double check that there is no existing approximate algorithm for this problem that would compare in efficiency with your algorithm but which would however not generalize to arbitrary distribution (e.g., various primal heuristics, moving between permutations by local changes)? In other words, is the strength of the proposed approach only in its generalizability to arbitrary distributions, or already for the distributions from $P_2^n(R^d)$?

Q2) In particular, the experiments do not convince me that min-SWGG is more efficient than Sinkhorn. The right-most graph in Figure 3 does not make clear to what accuracy the individual algorithms were run - in other words, it mixes accuracy and runtime (it does not compare the runtimes for the same achieved accuracy, or vice versa). Note, Sinkhorn keeps only dual variables and it need not keep the cost matrix (n-by-n) explicitly, so its memory complexity scales up well with n and d. Please comment.

Q3) Please comment on parallelizability of the algorithm. I guess that the gradient descent to minimize (9) is not easy to parallelize. Of course, random search is.

Q4) A suggestion: for small $d$, it might be interesting to try optimizing (9) (non-smoothed) by derivative-free methods, such as Nelder-Mead.

**Limitations:**

The experiments are done only for unweighted discrete distributions.

The experiments do not convincingly show that min-SWGG is more efficient than, e.g., Sinkhorn.

---

> ### Author Rebuttal · Authors · 2023-08-09
>
> We thank you for your work in revising our paper and for the valuable comments and questions.
>
> - **The formulation of min-SWGG is straightforward**. We respectfully disagree. You are correct in stating that PWD and min-SWGG share some similarities. While the definition of min-SWGG might appear simple in eq. (9), it can also be reformulated thanks to Wasserstein Generalized Geodesics (eq. 14), which is far from being trivial. Additionally, we want to emphasize that min-SWGG's appeal lies in two key properties: i) its ability to provide a transport plan, and ii) its suitability for optimization via gradient descent. These two properties do not hold with the original PWD loss.
>
> - **The color transfer experiment lacks sufficient details.** Details on the experiments can be found in [1]. More details will be added to the supplementary material.
>
> - **Regarding the ICP experiment.** Tables 1 and 2 (of the rebuttal's pdf) provide additional results for the ICP experiment (with $n=500$ and $n=3000$) and show that min-SWGG performs favorably when considering square Frobenius or Chamfer distance.
>
> - **The transport plan has (at most) n+m-1 non-null values.** This is a general property of an OT plan (see [2], Proposition 3.4, for example). In our case, the transport plans are optimal between $\mu_i$ ($i=1,2$) and the pivot measure $\nu$. This ensures that they have at most $n+m-1$ non-null values.
>
> - **Existing approximations of the linear assignment problem.** We thank the reviewer for this question. To the best of our knowledge, there is no algorithm that solves approximately (with guarantees) the linear assignment (Monge) problem with a complexity lower than quadratic (compared to our superlinear one), without additional assumptions about the nature of the problem/distributions. A relevant reference can be found in [3].  The reviewer is also correct in stating that the strength of the proposed approach also lies in its generalization to arbitrary distributions.
>
> - **Comparison with the Sinkhorn algorithm.** In the right part of Figure 3, we ran all the algorithms until convergence. Specifically, Sinkhorn was computed in a favorable convergence setup ($\epsilon=1$). Indeed, Sinkhorn keeps in memory the dense matrix via the dual variables which reduces the memory cost of the method. min-SWGG is more memory-efficient than Sinkhorn as it only stores the permutations ($n$ values). In the case of $\mu_1 \in \mathcal{P}_2^{n_1}(\mathbb{R^d})$ and $\mu_2 \in \mathcal{P}_2^{n_2}(\mathbb{R^d})$ with $n_1\neq n_2$, min-SWGG requires to store the quantile function, which consists of at most $n+m-1$ values.
>
> - **Parallelization of SWGG.** The reviewer is right. In terms of parallelization min-SWGG is comparable to the Sliced Wasserstein methods and may be parallelizable for random search but not for the optimization scheme.
>
>
> - **Gradient-free and Nelder-Mead methods.** We appreciate the reviewer's suggestion. We believe that gradient-free optimization schemes for min-SWGG could be practically beneficial, and we will closely examine them in our future investigations. However, let us note that the Nelder-Mead method requires the function to optimize to be continuous, which is not the case in our specific setting. Furthermore, when the measures belong to $\mathcal{P}_2^n(\mathbb{R^d})$, we are currently exploring differentiable sorting methods [4] that optimize directly over the permutahedron (the convex hull of permutations). These methods allow us to maintain the $O(n \log(n))$ complexity of the approach, and might be more efficient than the ad-hoc, yet simple, solution presented in our paper.*
>
> [1]Ferradans, S., Papadakis, N., Peyré, G., & Aujol, J. F. (2014). Regularized discrete optimal transport. SIAM Journal on Imaging Sciences, 7(3), 1853-1882.
>
> [2] Peyré, G., & Cuturi, M. (2019). Computational optimal transport: With applications to data science. Foundations and Trends® in Machine Learning, 11(5-6), 355-607.
>
> [3]Duan, R., & Pettie, S. (2014). Linear-time approximation for maximum weight matching. Journal of the ACM (JACM), 61(1), 1-23.
>
> [4]Blondel, M., Teboul, O. Berthet, Q. and Djolonga, J.  Fast Differentiable Sorting and Ranking, ICML (2020)

---

> > ### Comment · Reviewer_5WTB · 2023-08-15
> >
> > I thank the authors for their response. I like the paper and will be happy to see it published. Some   comments on comments:
> >
> > In Q1, I referred to linear assignment (LA) problem **with costs being squared Euclidean distances** (i.e., we have two sets of points and we seek to permute one of the sets so that the sum of squared distances of corresponding point paird is minimized), not just to the general LA problem. I am surprised that I did not hear about this problem before - it is so simple and natural! I wondered how much easier (in complexity) this problem is compared to the general LA problem and whether some classical approximation algorithms (other than for the general LA problem) exist. (I apologize that this question is not very objective.)
> >
> > Along the same line, it might be interesting to see how the (relatively complicated) machinery of generalized geodesics would simplify when we restrict the problem only to the above special case (i.e., with distributions from $P_2^n(R^d)$).

---

> > > ### Author Response · Authors · 2023-08-18
> > >
> > > We thank the reviewer for his/her positive assessment of our paper.
> > >
> > > The question raised by the reviewer ("Is there an efficient approximation of the Monge problem when the cost is the squared Euclidian distance ?") is indeed of interest and relevant for our problem. This Monge problem not only differs from a general LA problem by the fact that the considered graph is bipartite, but also because of the specific nature of the cost. It is at the heart of computational optimal transport problems. Let us start with the case where one of the two measures is absolutely continuous (*wrt.* the Lebesgue measure), it is possible to characterize the mapping as a gradient of a convex function, thanks to the Brenier Theorem. This had led to several popular approaches to approximate transport mapping with kernels for instance [1] or convex input neural network [2]. However, when the two measures are discrete and with the same number of atoms, one needs to rely on approximate solutions to avoid the super cubical complexity of an exact solver (based on Network simplex). Methods such as the Sinkhorn strategy discussed in the paper can be used [3]. Ad-hoc methods, such as hierarchical/multi-scale approaches [4], or convolution based [5], can be designed to lower the computational cost by leveraging the special structure of the squared Euclidian cost, but mostly rely on specific assumptions about the structure of the samples (*e.g.* regular grids) to be very efficient. In the specific case of multiscale methods [4], it is also unclear if one can simply backpropagate through this type of solvers, rendering it unpractical for machine learning applications.
> > >
> > > We will complete the related work of our paper with those missing references. Thanks again for your work and time on this paper.
> > >
> > >
> > > [1] Perrot M., Courty N., Flamary R., and Habrard A. (2016) Mapping Estimation for Discrete Optimal Transport. In NeurIPS, Barcelone, Spain
> > > [2] Makkuva, A., Taghvaei, A., Oh, S., & Lee, J. (2020). Optimal transport mapping via input convex neural networks. In ICML(pp. 6672-6681).
> > > [3] Altschuler J, Weed J, Rigollet P (2017) Near-linear time approximation algorithms for optimal transport via Sinkhorn iteration. In: Proceedings of NIPS 2017, pp 1961–1971
> > > [4] Schmitzer B (2016) A sparse multiscale algorithm for dense optimal transport. J Math Imaging Vis 56(2):238–259
> > > [5] Solomon J., de Goes F., Peyré G., Cuturi M., Butscher A., Nguyen A., Tao Du, and Guibas L. (2015) Convolutional wasserstein distances: efficient optimal transportation on geometric domains. ACM Trans. Graph. 34, 4, Article 66

---

### Author Rebuttal · Authors · 2023-08-09

We thank the reviewers for their constructive comments and answer their questions below. We hope our clarifications address the reviewers' questions.

Reviewers 5WTB and CJnA discussed the fact that the min-SWGG formulation is derived in the restricted $\mathcal{P}_2^n(\mathbb{R^d})$ setting and that the extension to the general $\mathcal{P}_2(\mathbb{R^d})$ setting is not clear. We propose a general answer to clarify this aspect.

We chose to derive the full article in the $\mathcal{P}_2^n(\mathbb{R^d})$ setting for sake of readability. Solving OT for generic 1D distributions is based on the generalized quantile function (as stated in A.2, see also [1], remark 2.30). Since min-SWGG relies on solving the 1D OT problem between the projected distributions (see eq. 8), it can be straightforwardly extended to generic distributions. In the final version, we propose to  make this more explicit by providing thorough details on the construction of the general version of min-SWGG (i.e. the setting where $\mu_1 \in \mathcal{P}_2^{n_1}(\mathbb{R^d})$ and $\mu_2 \in \mathcal{P}_2^{n_2}(\mathbb{R^d})$ with $n_1\neq n_2$), along with the corresponding code.

Additionally, we provide some figures and tables regarding the questions and propositions of the reviewers in the following pdf file.

Thank you again for your time and expertise put in reviewing this paper.

[1] Peyré, G., & Cuturi, M. (2019). Computational optimal transport: With applications to data science. Foundations and Trends® in Machine Learning, 11(5-6), 355-607.

---

### Decision · Program_Chairs · 2023-09-21

**Decision:**

Accept (spotlight)

**Comment:**

All reviewers viewed this work positively and highlighted the practical/theoretical value of the simple formulation here.  This paper is a clear "accept" and will find a clear audience at NeurIPS.

Please make sure the camera ready version of the paper addresses reviewer concerns and incorporates promised additions/changes from the rebuttal phase.